# Rethinking Layer Relevance in Large Language Models Beyond Cosine Similarity

**Cristian Hinostroza**[1,2], **Rodrigo Toro Icarte**[1,2], **Christ Devia**[2,3], **Andres Carvallo**[2],
**Eugenio Herrera-Berg**[2], **Denis Parra**[1,2], **Jorge F. Silva**[2,3]
[1]Pontificia Universidad Católica de Chile
[2]National Center for Artificial Intelligence (CENIA)
[3]Universidad de Chile

## Abstract

Large language models (LLMs) have revolutionized natural language processing. Understanding their internal mechanisms is crucial for developing more interpretable and optimized architectures. Mechanistic interpretability has led to the development of various methods for assessing layer relevance, with cosine similarity being a widely used tool in the field. In this work, we demonstrate that cosine similarity is a poor proxy for the actual performance degradation caused by layer removal. Our theoretical analysis shows that a layer can exhibit an arbitrarily low cosine similarity score while still being crucial to the model's performance. On the other hand, empirical evidence from a range of LLMs confirms that the correlation between cosine similarity and actual performance degradation is often weak or moderate, leading to misleading interpretations of a transformer's internal mechanisms. We propose a more robust metric for assessing layer relevance: the actual drop in model accuracy resulting from the removal of a layer. Even though it is a computationally costly metric, this approach offers a more accurate picture of layer importance, allowing for more informed pruning strategies and lightweight models. Our findings have significant implications for the development of interpretable LLMs and highlight the need to move beyond cosine similarity in assessing layer relevance.

## 1 Introduction

Transformers (Vaswani et al., 2017), initially designed for tasks related to large language models (LLMs) (Chkirbene et al., 2024), have become the main architecture for modern AI. They now support applications in computer vision (Caron et al., 2021), reinforcement learning (Li et al., 2023a), multimodal learning (Xu et al., 2023), recommender systems (Villa et al., 2020), and beyond. Since these models play a central role in AI, uncovering which parts matter the most can guide us toward more interpretable and optimized architectures.

*Mechanistic interpretability* aims to reverse-engineer pre-trained LLMs to better understand how they work (Ferrando et al., 2024). In this context, cosine similarity has become a standard tool for assessing semantic relationships between internal representations (Sanh et al., 2019; Li et al., 2023b; Sun et al., 2025; Modell et al., 2025). Intuitively, when the angle between two token embeddings is small, the tokens are assumed to encode similar information.

Recent studies have used cosine similarity to assess layer relevance in pre-trained LLMs (e.g., Sajjad et al., 2023; He et al., 2024; Men et al., 2025; Zhang et al., 2024b; Sun et al., 2025; Yang et al., 2024).The core idea is that layers making minimal changes to their input vectors are considered less relevant, with relevance quantified as one minus the cosine similarity between a layer's input and output vectors. This score has been applied in various contexts: for example, Gromov et al. (2025) used it to prune models and analyze performance across tasks, finding that reasoning tasks require more layers than factual ones. Similarly, He et al. (2024) visualized relevance scores across datasets (Figure 1B), showing that some layers consistently appear irrelevant regardless of the task.

While these results offer valuable insights, they hinge on the assumption that cosine similarity is a reliable indicator of layer relevance—an assumption we challenge. In this paper, we demonstrate that

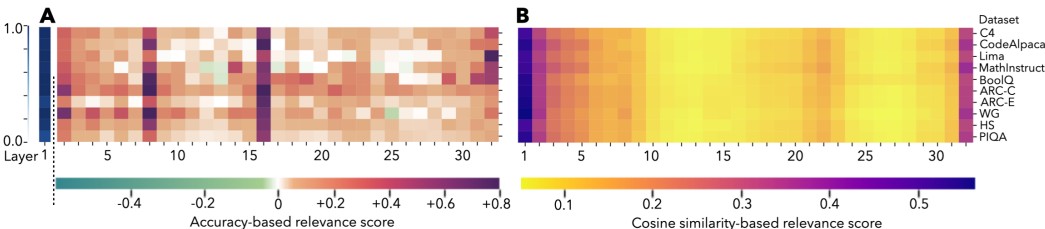

Figure 1: Relevance of OLMo (Groeneveld et al., 2024)'s layers across ten datasets. **A**. Accuracy-based relevance scores: We measure the drop in task accuracy to evaluate the relevance of each layer. Layers that increase accuracy when removed are highlighted in green, layers that do not affect accuracy appear in white, and layers that reduce the model's accuracy when removed are indicated in red/purple. **B**. Relevance scores computed using the cosine similarity score, which measures how much each layer transforms its input. The least relevant layers appear in yellow.

cosine similarity is a poor proxy for the actual performance degradation caused by layer removal. For example, layer 16 in OLMo appears to be of low relevance according to cosine similarity, as illustrated in Figure 1B (where irrelevant layers are shown in yellow). However, removing this layer results in an average accuracy drop of 66% across the ten datasets presented. In fact, eliminating layer 16 alone reduces OLMo's performance to chance level on ARC-C. These findings suggest that relying on cosine similarity as a relevance metric can lead to misleading interpretations of a transformer's internal mechanisms.

In this paper, we provide a formal proof demonstrating that a layer can exhibit an arbitrarily low cosine similarity score while still being crucial to the model's performance. In particular, removing such a layer can drastically alter the model's output—potentially reducing its accuracy from perfect to zero. This phenomenon arises when the layer introduces a subtle modification to its input vector that is subsequently amplified by downstream layers, resulting in a snowball effect. Consequently, despite its near-zero cosine similarity score, the removal of this layer can significantly disrupt the model's final predictions.

We then show that this theoretical worst-case scenario does occur, to some degree, in practice. Empirically, we find that the correlation between cosine similarity and actual performance degradation is often weak or moderate, depending on the model. As a result, cosine similarity either overestimates or underestimates a layer's true relevance in over 90% of cases we studied.

Having established that cosine similarity is an unreliable metric for assessing layer relevance, we next investigate the implications of re-running previously proposed experiments using a more robust alternative. Specifically, we argue that for the purposes of mechanistic interpretability, the most appropriate metric is the actual drop in model accuracy resulting from the removal of a layer. While this approach is computationally expensive—requiring layer-by-layer removal and performance re-evaluation—it avoids the shortcomings inherent to cosine similarity. Crucially, this metric captures the complex interdependencies among layers in Transformer architectures.

We begin by replicating the layer relevance visualization introduced by He et al. (2024). Figure 1A displays the relevance of each layer in OLMo across ten datasets, measured by the change in model accuracy after removing each layer individually. Red/purple indicates a drop in accuracy, green an improvement, and white no change. This visualization offers a markedly different perspective from cosine similarity, revealing that layer relevance varies by dataset and highlighting the critical role of layers 8 and 16 in OLMo's performance.

We then replicated the task analysis proposed by Gromov et al. (2025), which involved pruning layers deemed irrelevant based on cosine similarity and observing the resulting performance drop. Instead, we ranked layers by the actual decrease in accuracy on the task's training set and pruned accordingly. Results are shown in Figure 2. Because our metric better reflects layer relevance, the performance drop in HellaSwag is less pronounced than in the original study. This challenges the conclusion that all layers are essential for reasoning tasks: using a more informative metric, we find that over 75% accuracy can be maintained even after removing 22% of the layers.

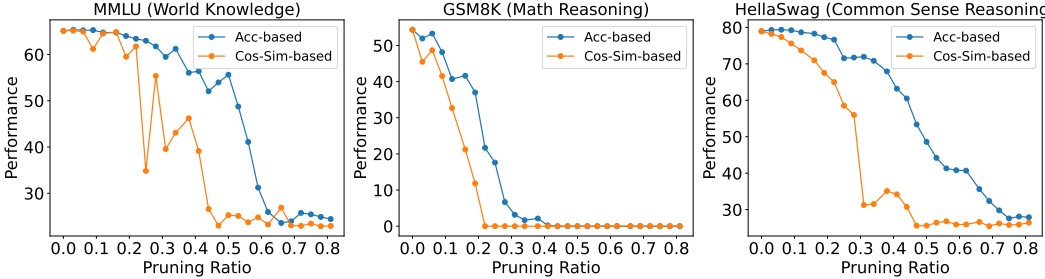

Figure 2: We evaluate LLaMA-3-8B using the cosine-similarity pruning strategy proposed by Gromov et al. (2025), and compare it with our method. In contrast to cosine similarity, our approach mitigates immediate performance degradation in reasoning tasks, highlighting the critical role of selecting an appropriate metric for interpreting model internals.

We conclude with a practical application in structured pruning (Anwar et al., 2017), which aims to remove layers from trained models with minimal impact on performance. In the task-dependent setting, we show that pruning layers based on our accuracy-based relevance score yields superior results compared to existing methods, including Taylor approximations (Kim et al., 2024; Ma et al., 2023), cosine similarity (He et al., 2024; Men et al., 2025; Gromov et al., 2025), and FinerCut (Zhang et al., 2024b). In the task-independent setting, our method also achieves the best performance, though it is sensitive to the choice of calibration dataset.

## 2 RELATED WORK

A central challenge in Transformer research is accurately measuring layer relevance. This question is critical for two main applications: mechanistic interpretability, which seeks to understand how pre-trained LLMs operate, and structured pruning, which aims to reduce model size by removing irrelevant layers while preserving performance. Cosine similarity has become a popular metric for both tasks due to its computational efficiency and intuitive appeal (e.g., Sajjad et al., 2023; Gromov et al., 2025; He et al., 2024; Men et al., 2025; Yang et al., 2024; Zhang et al., 2024b). It assumes that layers making minimal changes to their input vectors are less relevant. Moreover, cosine-based pruning has achieved strong results in task-independent settings.

However, cosine similarity is only a proxy for what truly matters: downstream performance. While prior work has raised concerns about its use in comparing token embeddings (Timkey & Van Schijndel, 2021), to our knowledge, this is the first study to rigorously evaluate—both theoretically and empirically—its limitations in estimating layer relevance in Transformer models. We then propose an alternative: an accuracy-based relevance score, which considers a layer relevant only if its removal significantly degrades performance on a given task.

Beyond cosine similarity, which is typically used as a local metric (e.g., Sajjad et al., 2023; Gromov et al., 2025; He et al., 2024; Men et al., 2025), several global metrics have been proposed. These assess relevance by evaluating changes in the model's output after removing a layer. Global metrics fall into two categories: consistency-based and performance-based. Consistency-based metrics compare the model's output distributions with and without a target layer (Sieberling et al., 2024; Yang et al., 2026; Zhang et al., 2024b), identifying layers whose removal leaves the output unchanged. However, these metrics focus on output invariance rather than predictive accuracy, and may overlook layers that subtly affect performance.

Performance-based metrics are more aligned with our approach (Kim et al., 2024; Ma et al., 2023; Zhong et al., 2025; Song et al., 2024). These metrics rely on ground-truth information to assess the relevance of a layer. For example, Ma et al. (2023) use Taylor expansions to estimate the change in loss when a layer is removed. Other works rely on perplexity-based scores, deeming layers irrelevant if their removal does not significantly increase perplexity (Kim et al., 2024; Zhong et al., 2025; Song et al., 2024). Like our accuracy-based score, these methods aim to identify layers whose exclusion yields minimal performance degradation. Nevertheless, as we show in Section 6, our metric consistently outperforms these alternatives in structured pruning tasks.

Finally, our work connects with a broader literature on understanding how Transformers represent and process information (Brinkmann et al., 2024; Clark et al., 2019b; Devlin et al., 2019; Geva et al., 2021; 2022; 2023; Gurnee & Tegmark, 2024; Jawahar et al., 2019; Lioubashevski et al., 2025; Meng et al., 2022; Sun et al., 2025; Tigges et al., 2023). In particular, our relevance metric could be used to revisit studies that identify functional behaviors in specific attention layers (Clark et al., 2019b; Geva et al., 2023) or MLPs (Geva et al., 2021; Meng et al., 2022), offering new insights into their contributions. It may also help bridge findings on global Transformer behavior (Gurnee & Tegmark, 2024; Brinkmann et al., 2024; Tigges et al., 2023) with specific layers or processing stages. We expand on these connections and review additional pruning methods in Appendix A.

## 3 COSINE-SIMILARITY SCORE

Let us begin by formally defining the *cosine-similarity score*. The cosine-similarity score is a local metric that examines the difference between the input and output vectors of a layer to assess its relevance (Sajjad et al., 2023; Gromov et al., 2025; He et al., 2024; Men et al., 2025). Intuitively, if the output of a layer is identical to its input, removing that layer would have no effect on the model's performance. Formally, given two vectors $\boldsymbol{x}$ and $\boldsymbol{y}$, the cosine similarity is defined as follows:

$$\text{CosineSim}(\boldsymbol{x}, \boldsymbol{y}) = \frac{\boldsymbol{x} \cdot \boldsymbol{y}}{||\boldsymbol{x}|| \cdot ||\boldsymbol{y}||} \tag{1}$$

To define a score where the least relevant layers receive a value of zero, we compute the cosine-similarity score as one minus the cosine similarity between the input and output vectors of a layer. Given a calibration dataset $\mathbb{D} = \{s^{(i)}\}_{i=1}^{N}$, the relevance of a layer is then calculated as the average cosine-similarity score across all tokens and instances:

$$\text{CosSimScore}(l; \mathbb{D}) = \frac{1}{N} \sum_{i=1}^{N} \frac{1}{n^{(i)}} \sum_{j=1}^{n^{(i)}} \left( 1 - \text{CosineSim}\big(\boldsymbol{X}_{j,:}^{(l,i)}, \boldsymbol{X}_{j,:}^{(l+1,i)}\big) \right), \tag{2}$$

where each sequence $s^{(i)}$ has $n^{(i)}$ tokens, $\boldsymbol{X}^{(l,i)} \in \mathbb{R}^{n^{(i)} \times d}$ is the intermediate layer representation of $s^{(i)}$ at layer $l$, and $\boldsymbol{X}_{j,:}^{(l,i)} \in \mathbb{R}^d$ denotes the representation of the $j$-th token at layer $l$.

## 4 RETHINKING LAYER RELEVANCE: BEYOND COSINE SIMILARITY

This section highlights the limitations of cosine similarity as a layer relevance metric. We show, both theoretically and empirically, that layers assessed as irrelevant by cosine similarity can still cause significant drops in downstream performance when removed. To address this, we propose an accuracy-based metric that directly evaluates relevance based on what truly matters: the model's predictive performance.

### 4.1 LIMITATIONS OF COSINE SIMILARITY FOR LAYER RELEVANCE

We begin by formally demonstrating that a layer can have an arbitrarily low cosine similarity score while still having a significant impact on model performance. Specifically, the following theorem shows that for any dataset $\mathbb{D}$ and any $\epsilon > 0$, it is possible to construct a decoder-only Transformer that achieves perfect accuracy on $\mathbb{D}$, yet the removal of the layer with the lowest cosine similarity reduces the model's performance to zero. Moreover, the cosine similarity score of that layer is $\epsilon$.

**Theorem 1** *Let $f^L$ denote a Transformer model with $L$ layers, and $f_{-l}^L$ represent the same model with layer $l$ removed. Then, for any $\epsilon > 0$ and any calibration dataset $\mathbb{D} = \{(s^{(i)}, y^{(i)})\}_{i=1}^{N}$ such that $s^{(i)} \neq s^{(j)}$ for all $i \neq j$ and $y^{(i)} \in \{0, \dots, C-1\}$, there exists a decoder-only Transformer $f^L$ with $L \geq 3$ satisfying the following conditions:*

1. *There exists an intermediate layer $l \in \{1, \dots, L-2\}$ such that $\text{CosSimScore}(l; \mathbb{D}) = \epsilon$, and $\text{CosSimScore}(i; \mathbb{D}) > \epsilon$ for all $i \neq l$.*

2. *The full model achieves perfect accuracy: $f^L(s^{(i)}) = y^{(i)}$ for all $s^{(i)} \in \mathbb{D}$, but removing layer $l$ causes the model's accuracy to drop to zero: $f_{-l}^L(s^{(i)}) \neq y^{(i)}$ for all $s^{(i)} \in \mathbb{D}$.*

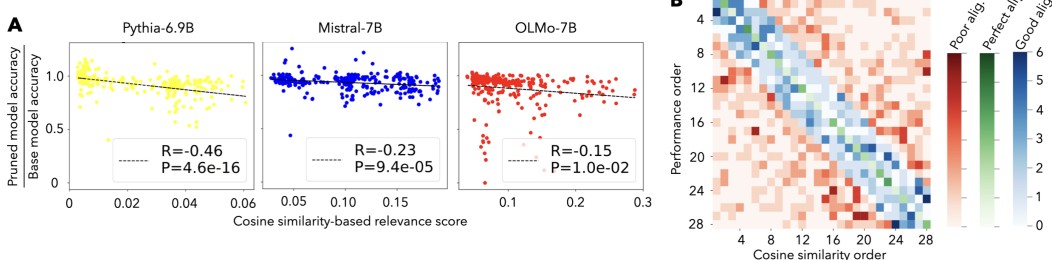

Figure 3: **A.** Relationship between cosine similarity scores and performance variation after removing a layer. Each point represents a specific layer–task pair from one of the 28 middle layers in Pythia, Mistral, or OLMo, evaluated across ten tasks (same set as in Figure 1). **B.** Alignment between cosine similarity rankings and performance rankings across three models, ten tasks, and 28 layers. Cell $(i, j)$ indicates the number of times cosine similarity assigned rank $j$ while the ground-truth rank was $i$ (rank 1 = least relevant). The heatmap uses three distinct color scales: green for the diagonal (perfect alignment), blue for low-cost misrankings, and red for all other cells.

To construct a Transformer in which a layer has an arbitrarily low cosine similarity yet significantly impacts model performance, two key conditions must be met. First, a snowball effect must occur: the target layer introduces a subtle change to its input vector, which is then amplified by subsequent layers. This allows the layer to have minimal cosine similarity while still influencing the final output. Second, some embedding dimensions must be irrelevant to the model's prediction. This enables other layers to make large changes in those irrelevant dimensions, inflating their cosine similarity scores without contributing to performance. A complete proof is provided in Appendix B.

We believe both phenomena can naturally arise in pre-trained LLMs, particularly in task-dependent settings. For a given task, many transformations applied by the model may be irrelevant to solving it. Empirically, we observe a snowball effect in models like OLMo, where layer 16 exhibits a very low cosine similarity score yet has a substantial impact on performance (see Figure 1).

We now present a more in-depth empirical evaluation of the cosine similarity score as a proxy for layer relevance. Specifically, we aim to assess how well cosine similarity predicts the actual drop in downstream performance when a layer is removed. Figure 3A compares the cosine similarity score with the observed reduction in accuracy after removing individual layers from three pre-trained LLMs—Mistral, Pythia, and OLMo—across ten datasets: C4, CodeAlpaca, LIMA, MathInstruct, BoolQ, ARC-Challenge, ARC-Easy, HellaSwag, PIQA, and Winogrande. We exclude the first and last two layers, as they are trivially identifiable as relevant and behave as clear outliers.

As shown in Figure 3A, there is some correlation between cosine similarity and performance degradation. However, the strength of this correlation varies by model: moderate in Pythia (R = -0.46), weak in Mistral (R = -0.23), and very weak in OLMo (R = -0.15).

To further evaluate the reliability of cosine similarity, we compare its layer relevance ranking against a ground-truth ranking based on actual performance drop. Figure 3B presents a confusion matrix summarizing the results across the same three models and ten datasets. In this matrix, cell $(i, j)$ indicates the number of times cosine similarity ranked a layer as the $j$-th least relevant, while its true rank was $i$ according to performance drop. Diagonal entries represent perfect agreement; entries below the diagonal indicate underestimation of relevance, and those above indicate overestimation. Overall, cosine similarity misestimated a layer's relevance in 93.8% of cases. That said, not all errors are equally severe: entries near the diagonal reflect small ranking deviations. But even when considering only substantial errors (highlighted in red), cosine similarity still fails in 53.6% of cases.

Overall, these results demonstrate that cosine similarity is an unreliable and noisy metric for estimating layer relevance. In many cases, layers deemed irrelevant by cosine similarity lead to substantial drops in performance when removed—and vice versa. This inconsistency highlights the need for caution when using cosine similarity, particularly in the context of mechanistic interpretability. Relying on such a flawed metric risks drawing incorrect conclusions about how Transformer models function. We illustrate this issue with two concrete examples in Section 5.

## 4.2 ACCURACY-BASED RELEVANCE SCORE

Rather than relying on a proxy, we propose directly visualizing the performance drop to assess how each layer contributes to the model's effectiveness. We do so by using an *accuracy-based score*. Given a dataset $\mathbb{D}$ and a Transformer model with $L$ layers $f^L$, we assess the relevance of layer $l$ as:

$$\text{AccBasedRelevance}(f^L, l, \mathbb{D}) = 1 - \frac{\max(\text{Accuracy}(f^L_{-l}, \mathbb{D}) - r(\mathbb{D}), 0)}{\max(\text{Accuracy}(f^L, \mathbb{D}) - r(\mathbb{D}), 0)}, \quad (3)$$

where $\text{Accuracy}(f^L, \mathcal{D})$ denotes the accuracy of the full model on dataset $\mathbb{D}$, and $r(\mathbb{D})$ represents the expected performance of a random predictor in the dataset.

This score ranges from $-\infty$ and $+1$: negative values indicate improved performance upon removal of the layer, zero indicates no change, and positive values reflect a drop in performance. Thus, higher scores correspond to greater relevance of the layer for the task. It is important to note that this range is valid only when the full model performs better than a random predictor. If the model's accuracy falls below that of a random predictor, the relevance score becomes ill-defined, and the analysis should not be applied in such cases.

The accuracy-based score can be applied to any component of a transformer-based model, including a single weight, a multi-head attention layer, an MLP, a Transformer block, or multiple blocks. That said, in the next section, we will focus on visualizing the importance of Transformer blocks.

## 5 CASE STUDIES

To assess the practical impact of our findings, we revisit two case studies that used cosine similarity to evaluate layer relevance in pre-trained LLMs. Replacing cosine similarity with our accuracy-based metric, we observed significantly different outcomes. These results highlight the limitations of proxy metrics and reinforce the value of accuracy-based evaluation for mechanistic interpretability.

## 5.1 RELEVANCE CONSISTENCY ACROSS DATASETS

We begin by revisiting the study *What Matters in Transformers* by He et al. (2024), which proposes a method to visualize layer relevance using cosine similarity. Figure 4B shows the relevance of each layer in Mistral (Jiang et al., 2023) across multiple datasets, with yellow indicating low cosine similarity score and purple indicating high. Based on these visualizations, the authors conclude that layer relevance is largely task-independent—a pattern we also observed in OLMo (Figure 1B).

When applying our accuracy-based metric, we obtain a markedly different view of layer relevance, as shown in Figure 1A for OLMo and Figure 4A for Mistral. These visualizations use a fixed color scale: green for performance gains, white for no change, and red/purple for performance drops. Unlike cosine similarity, our metric reveals that layer relevance is highly task-dependent. For example, removing block 14 in OLMo reduces accuracy by ∼41% on MathInstruct but has minimal impact (∼1%) on CodeAlpaca. Some layers even show negative relevance, improving performance when removed—e.g., block 23 in Mistral increases accuracy by ∼25% on MathInstruct but decreases it by ∼6% on CodeAlpaca. Finally, our metric also captures broader task sensitivity. For instance, Mistral shows consistently lower relevance across blocks on MMLU compared to BoolQ—a distinction not visible in cosine similarity plots.

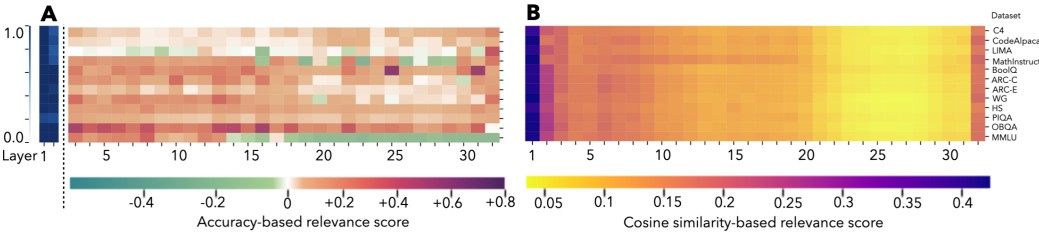

Figure 4: Relevance of Mistral's Transformer blocks across datasets. **A**. Accuracy-based relevance scores. **B**. Cosine similarity score.

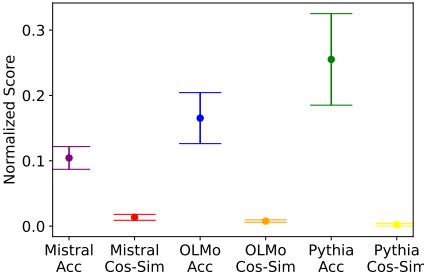

Figure 5: Relevances Across Datasets

To ensure these differences are not artifacts of visualization, we conducted a statistical comparison between the two metrics. Using z-score normalization, we computed the average variance of each of OLMo's 32 blocks across ten datasets. As shown in Figure 5, our accuracy-based score exhibits significantly greater variance than cosine similarity. A Wilcoxon test (Appendix C.3) confirms these differences are statistically significant, reinforcing the visual evidence that layer relevance is task-dependent.

Beyond cross-task consistency, we also explored how relevance evolves during training (Appendix C.4) and pruning (Appendix C.5). In pruning, we found that a layer's relevance depends on the presence of other layers—removing one can increase or decrease the importance of another. In training, no clear pattern emerged: some layers gained relevance over time, while others fluctuated.

## 5.2 DIFFERENCES BETWEEN TYPES OF TASKS

We now revisit *The Unreasonable Ineffectiveness of the Deeper Layers* by Gromov et al. (2025), which argues that deeper layers in pre-trained LLMs are essential for reasoning tasks (e.g., GSM8K, HellaSwag) but less relevant for factual retrieval tasks such as MMLU. Their hypothesis is based on the idea that, when faced with a reasoning task, the model must compute intermediate steps to arrive at the final answer—implying that all layers contribute meaningfully to such tasks. Their analysis on LLaMA 2-70B showed that MMLU retained accuracy under early pruning, while GSM8K and HellaSwag degraded instantly and continued to decline, supporting the hypothesis that deeper layers play a critical role in reasoning.

Their pruning strategy, however, was based on cosine similarity rather than direct depth-based ablation. To assess the robustness of their findings, we replicated the experiment on LLaMA 3-8B using our accuracy-based relevance metric. As shown in Figure 2, we observed similar task-dependent trends according to the cosine similarity score: MMLU remained stable under initial pruning, while GSM8K and HellaSwag showed performance drops, particularly in GSM8K.

In contrast, when pruning is guided by our accuracy-based metric, a different pattern emerges: the model maintains strong performance on HellaSwag even after several blocks are removed, and GSM8K shows minimal degradation after pruning two blocks. This suggests that cosine similarity may underestimate the importance of certain blocks for reasoning tasks—pruning them prematurely and thereby reducing model performance. Moreover, cosine similarity appears unable to identify blocks that are not relevant for reasoning, whereas our method successfully distinguishes between essential and non-essential layers.

Finally, we note that, unlike Gromov et al. (2025), who pruned contiguous block groups based on aggregate cosine similarity, our method prunes blocks iteratively, re-evaluating the model after each step. In Appendix D, we compare both strategies—cosine similarity as used in their work and our iterative method. While some trends are less pronounced, the core conclusion remains: different metrics yield different insights into model behavior.

## 6 EMPIRICAL RESULTS IN STRUCTURED PRUNING

The primary goal of our accuracy-based relevance score was to support mechanistic interpretability by providing a reliable measure of layer importance. However, this metric also proves effective for structured pruning—i.e., reducing model size by removing layers with minimal impact on performance (Anwar et al., 2017). Surprisingly, pruning layers deemed irrelevant by our score yields state-of-the-art results, while remaining simple to implement.

Structured pruning methods typically rely on a calibration set to estimate layer relevance and prune up to $p\%$ of the model's weights. These methods differ in whether they apply one-shot or iterative pruning, and in the criteria used to rank layers. To evaluate pruning effectiveness, we compare generalization performance across standard benchmarks. We applied our accuracy-based score iteratively to prune LLaMA3-8B (Grattafiori et al., 2024), selected for its use in prior state-of-the-art

Table 1: Task-dependent results for LLaMA3-8B models across multiple tasks. All methods remove 25% of the model using each task's training set. "Original" refers to the unpruned model.

| Method | Arc-C | Arc-E | BoolQ | HS | OBQA | PIQA | WG | MMLU | Mean |
|---|---|---|---|---|---|---|---|---|---|
| Original | 53.16 | 81.02 | 82.02 | 78.94 | 44.8 | 81.28 | 73.56 | 65.11 | 69.99 |
| Taylor | 31.48 | 67.97 | 61.31 | 62.73 | 38.4 | 76.55 | 55.64 | 25.03 | 52.39 |
| Cosine Similarity | 45.73 | 67.8 | 66.33 | 69.52 | 38.6 | 72.91 | 71.35 | 44.05 | 59.54 |
| Out. Cosine-Sim | 39.51 | 65.57 | 72.11 | 67.97 | 36.8 | 76.88 | 65.11 | 36.82 | 57.6 |
| Out. Norm-Sim | 40.19 | 66.08 | 72.08 | 64.96 | 39.6 | 75.46 | 68.27 | 49.03 | 59.46 |
| Out. Divergence-Sim | 41.13 | 65.87 | 72.14 | 67.20 | 34.0 | 74.43 | 69.46 | 35.12 | 57.42 |
| Perplexity | 38.14 | 53.11 | 62.14 | 58.92 | 38.4 | 67.19 | 62.12 | 59.04 | 54.88 |
| Slice-GPT | 41.64 | 73.27 | 75.75 | 67.35 | 39.6 | 77.15 | 70.56 | 48.74 | 61.76 |
| Accuracy (Ours) | **49.57** | **74.96** | **84.04** | **71.53** | **44** | **79.06** | **73.8** | **62.97** | **67.49** |

pruning work (Zhang et al., 2024b). We also replicated the experiment on Mistral-7B (Jiang et al., 2023) and evaluated one-shot pruning (see Appendix E.3).

Our method was benchmarked against leading pruning techniques, including: Taylor approximations (Kim et al., 2024; Ma et al., 2023), cosine similarity (He et al., 2024; Men et al., 2025; Gromov et al., 2025), output-based metrics (e.g., output cosine similarity, norm similarity, divergence similarity) (Zhang et al., 2024b; Yang et al., 2026; Sieberling et al., 2024), and perplexity-based relevance (Kim et al., 2024; Zhong et al., 2025; Song et al., 2024).

To ensure fair comparison, all methods pruned the same layer types using identical calibration data, removing up to 25% of the model. Metrics were recomputed after each pruning step. We also included SlideGPT (Ashkboos et al., 2024), which reduces layer size rather than removing entire layers; we matched its pruning ratio to 25%. No healing or postprocessing was applied, as our focus was on evaluating the effectiveness of the relevance metric itself.

We assessed performance across eight widely used benchmarks: ARC-Challenge (Clark et al., 2018), ARC-Easy (Clark et al., 2018), BoolQ (Clark et al., 2019a), HellaSwag (Zellers et al., 2019), PIQA (Bisk et al., 2020), OpenBookQA (Mihaylov et al., 2018), Winogrande (Sakaguchi et al., 2021), and MMLU (Hendrycks et al., 2021). These span a range of reasoning and knowledge tasks, each with a train/test split. Implementation details are provided in Appendix E.1.

We first report task-dependent results, where the goal is to optimize performance for a specific task. Each model was pruned using the corresponding training set as calibration data and evaluated on the test set. Table 1 presents results for LLaMA3-8B. Our accuracy-based score consistently outperformed all baselines and, in some cases, even surpassed the unpruned model. Similar trends were observed with Mistral-7B (see Appendix E.2). These findings indicate that our score can effectively prune pre-trained LLMs when the deployment task is known. For example, if a lightweight model is needed for math problem solving, our score identifies and removes layers unrelated to that domain. While this may reduce performance on unrelated tasks (e.g., poetry generation), such trade-offs are acceptable when the goal is task-specific efficiency.

We now turn to the task-independent setting, where the objective is to prune a pre-trained LLM while preserving performance across a diverse set of tasks. In this context, we observed that the effectiveness of our accuracy-based score is highly sensitive to the choice of calibration set.

Table 2 reports results for LLaMA3-8B using a calibration set composed of 10% of the training data from each of the eight benchmarks. Under this configuration, our method outperforms all baselines, yielding a pruned model that achieves the highest average performance across tasks. However, when the calibration set is restricted to a single benchmark, performance varies significantly (as shown in Appendix E.5) while cosine similarity remains stable at $\approx 60\%$, regardless of the calibration set.

To further examine our model's sensitivity to the choice of calibration set, the last two rows of Table 2 report its task-independent performance when pruned using two different calibration sets: ARC-E and C4. First, pruning with ARC-E yields strong performance across most tasks, indicating that layer relevance derived from one task can generalize effectively to others. Second, pruning with C4 demonstrates the opposite effect: task-specific relevance can severely degrade performance on

Table 2: Task-independent results for LLaMA3-8B across multiple tasks. Each pruning method uses the same calibration dataset to prune 25% of the model once, which is then evaluated on all tasks.

| Method | Arc-C | Arc-E | BoolQ | HS | OBQA | PIQA | WG | MMLU | Mean |
|---|---|---|---|---|---|---|---|---|---|
| Original | 53.16 | 81.02 | 82.02 | 78.94 | 44.8 | 81.28 | 73.56 | 65.11 | 69.99 |
| Taylor | 45.39 | 67.97 | 61.31 | 62.73 | 41.4 | 76.55 | 68.11 | 25.03 | 56.06 |
| Cosine Similarity | 43.6 | 66.96 | 75.53 | 69.35 | 36.2 | 73.23 | **71.82** | 44.07 | 60.1 |
| Out. Cosine-Sim | 40.61 | 65.78 | 67.58 | 64.6 | 36.2 | 75.3 | 69.51 | 30.16 | 56.22 |
| Out. Norm-Sim | 37.88 | 65.32 | 57.77 | 61.73 | 39.6 | 74.86 | 65.43 | 26.5 | 53.64 |
| Out. Divergence-Sim | 39.51 | 64.39 | 64.8 | 64.37 | 34.2 | 73.83 | 68.59 | 33.5 | 55.4 |
| Perplexity | 31.83 | 48.95 | 59.27 | 48.01 | 30.5 | 66.76 | 61.88 | 29.31 | 47.06 |
| Slice-GPT | 41.16 | 70.28 | 77.49 | 61.19 | 36.8 | 73.66 | 62.66 | 45.03 | 58.53 |
| Accuracy (Ours) | 47.35 | 71.68 | **78.38** | 73.41 | **43.8** | 76.55 | 71.11 | **58.04** | **65.04** |
| Accuracy (Arc-E) | **51.37** | **74.96** | 66.94 | **73.62** | 43.6 | **78.51** | 71.59 | 44.82 | 63.18 |
| Accuracy (C4) | 36.69 | 56.52 | 53.36 | 60.16 | 33.8 | 72.63 | 60.14 | 28.5 | 50.23 |

unrelated tasks. Finally, certain tasks remain difficult to generalize to. In particular, BoolQ and MMLU exhibit relevance patterns that differ substantially from the rest, such that strong performance was only achievable when incorporating part of their training data during pruning. Overall, these findings reinforce a key observation of our work: layer relevance is inherently task-dependent.

An important direction for future work is to understand what properties a calibration set should possess to enable strong task-independent performance. Our observations suggest that accuracy improves when the calibration set includes a diverse mix of data—for example, sampling approximately 10% from the training sets of multiple benchmarks. We also find that certain tasks, such as ARC-E, exhibit layer relevance patterns that align well with many other tasks. However, the underlying reasons for this consistency remain unclear and warrant further investigation.

## 6.1 COMPUTATIONAL COST COMPARISON

As discussed in Section 4, cosine similarity has clear limitations as a measure of layer relevance. Its primary advantage, however, is speed. Computing layer relevance via cosine similarity is highly efficient: let $N$ denote the number of layers and $T$ the number of instances in the calibration set. Cosine similarity requires only $T$ forward passes to compute relevance for all layers. In contrast, our accuracy-based score requires $N \times T$ forward passes, output-based methods require $(N + 1) \times T$ forward passes, and Taylor approximations require $T$ forward and $T$ backward passes.

Table 3 illustrates this trade-off by reporting the time required to prune 25% of LLaMA-3-8B. As shown, cosine similarity is by far the fastest—requiring only a few minutes to prune 25% of the model. Our accuracy-based method averages 4.6 hours, which is similar to other baselines. However, our method consistently produces pruned models with superior performance.

Reducing the computational cost of our metric is a key direction for future work, particularly for larger models. Using our current (naïve) implementation, we estimate that pruning 50% of LLaMA-3-70B would take between 8.5 days with C4 and 1.1 days with CodeAlpaca—using two NVIDIA H100 GPUs. These estimates represent worst-case scenarios, as we have not yet fully exploited parallelization. Further details are provided in Appendix E.6.

Moreover, our iterative pruning strategy is inherently suboptimal. At present, we employ a greedy approach: we compute each layer's relevance, prune the least relevant layer, and then recompute relevance scores before proceeding to the next pruning step. However, identifying the optimal combination of $n$ layers to remove would require a search-based method—such as A*—capable of backtracking and considering cases where removing a seemingly suboptimal layer might lead to better overall performance later. Unfortunately, even our current greedy procedure is computationally expensive, making exhaustive search for the optimal set of $n$ layers infeasible. Nevertheless, if we could accelerate the evaluation (or estimation) of accuracy-based relevance, it would open the door to exploring how much performance could be gained by finding truly optimal pruning combinations.

Table 3: End-to-end runtime for pruning 25% of LLaMA-3-8B on NVIDIA L40s GPU.

|  | Accuracy | Cos. Sim. | Perplexity | Out. Cos | Out. Norm | Out. Div | Taylor |
|---|---|---|---|---|---|---|---|
| C4 | 7.70 hrs | 9.54 min | 7.71 hrs | 8.08 hrs | 7.98 hrs | 8.08 hrs | 11.30 hrs |
| LIMA | 6.46 hrs | 14.45 min | 6.38 hrs | 6.62 hrs | 6.65 hrs | 6.83 hrs | 14.17 hrs |
| MathInstruct | 3.16 hrs | 7.01 min | 3.30 hrs | 3.22 hrs | 3.29 hrs | 3.31 hrs | 8.45 hrs |
| CodeAlpaca | 1.06 hrs | 3.55 min | 1.08 hrs | 1.12 hrs | 1.11 hrs | 1.12 hrs | 7.59 hrs |
| Mean | 4.60 hrs | 8.64 min | 4.62 hrs | 4.76 hrs | 4.76 hrs | 4.83 hrs | 10.37 hrs |

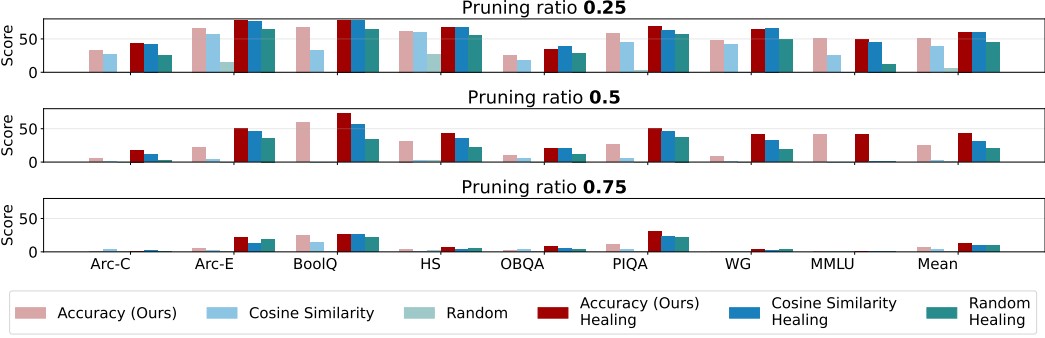

Figure 6: Impact of healing (dark colors) after pruning (light colors) across varying pruning ratios.

## 6.2 RESULTS OF PRUNING WITH HEALING

Recent work has introduced a healing phase after pruning (e.g., Sun et al., 2024; Gromov et al., 2025; Song et al., 2024; Kim et al., 2024). This phase fine-tunes the pruned model for a few steps, aiming to mitigate distributional mismatches across layers caused by pruning. In this section, we examine whether healing complements our accuracy-based pruning approach.

Figure 6 shows task-dependent pruning results with and without healing. Scores are normalized: 100 indicates perfect performance, 0 a random predictor. When pruning 25% of layers, performance after healing is nearly invariant to the pruning strategy. In fact, a random baseline—removing 25% of layers at random (averaged over five seeds)—performs competitively after healing. In other words, for small pruning ratios, healing largely neutralizes differences between pruning methods. Still, accuracy-based pruning combined with healing slightly outperforms cosine similarity (60.5% vs. 59.6% normalized average). At 50% pruning, the gap widens: accuracy-based + healing achieves 42.2% normalized performance, compared to 31.2% for cosine similarity and 20.5% for random. At 75% pruning, all methods degrade severely, yet accuracy-based pruning remains superior (12.2% vs. 9.3% for cosine similarity and 9.4% for random).

Implementation details are provided in Appendix E.7. In brief, we fine-tuned using LoRA for up to 10 epochs, reporting the best performance across epochs (full curves appear in Appendix E.7). Ten epochs are sufficient for all methods to exhibit signs of overfitting. While early stopping relied on the test set—ideally, a validation set should be used—this does not alter the main conclusion: healing substantially reduces differences between pruning strategies at low pruning ratios, yet accuracy-based pruning consistently achieves the best performance as pruning becomes more aggressive.

## 7 CONCLUSION

In this paper, we challenged the common use of cosine similarity as a proxy for layer relevance in LLMs, showing through theory and experiments that it often misrepresents true importance. To address this, we propose an accuracy-based relevance metric that directly measures performance impact, offering a more faithful view of layer significance. Beyond interpretability, this metric enables superior structured pruning, outperforming existing methods in both task-dependent and task-independent settings. Our findings call for a shift toward performance-grounded evaluations to better understand model internals and design more effective pruning strategies.

ACKNOWLEDGMENTS

We gratefully acknowledge the support of the Agencia Nacional de Investigación y Desarrollo (ANID), Chile. This research was funded by the National Center for Artificial Intelligence CENIA FB210017, Basal ANID. The work of R. Toro Icarte was supported by Fondecyt Iniciación 11230762. The work of C. Devia was supported by Fondecyt 11241551 and FONDEQUIP EQM230106. The work of A. Carvallo was supported by Fondecyt 3240001. The work of D. Parra was supported by Fondecyt Regular 1231724 and the Millennium Initiative research centers iHealth ICN2021_004 and IMFD ICN17_002. The work of J. F. Silva was supported by Fondecyt Regular 1250098 and ANID AC3E CIA250006.

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

# A RELATED WORK (EXTENDED VERSION)

## A.1 UNDERSTANDING TRANSFORMER INTERNALS

The question of how Transformer models represent and process information was first explored in depth with BERT (Devlin et al., 2019). Early studies revealed that BERT captures structural properties of language across its layers. Lower layers focus on phrase-level and surface features, while intermediate layers encode a rich hierarchy of linguistic information—starting with syntactic structures and transitioning to semantic representations at higher layers (Jawahar et al., 2019). Additionally, some attention heads within BERT specialize in specific linguistic tasks, such as syntactic parsing and coreference resolution, aligning with traditional linguistic notions (Clark et al., 2019b). These findings provided initial insights into how Transformer-based models organize knowledge, setting the stage for broader investigations into their internal mechanisms.

Recent studies have further refined our understanding of how Transformers encode and manipulate information. Feed-forward (FF) layers function as key-value memory systems, storing patterns from training and influencing the model's output distribution (Geva et al., 2021). This structured memory is particularly important for factual recall, as knowledge is primarily stored in the FF layers of middle blocks (Meng et al., 2022). Meanwhile, attention layers propagate and retrieve stored information, dynamically integrating relevant associations for prediction (Geva et al., 2023).

Beyond factual recall, Transformers encode abstract and structured representations. They capture spatiotemporal relationships in text (Gurnee & Tegmark, 2024) and implement a depth-bounded recurrent mechanism that stores intermediate results at selected token positions (Brinkmann et al., 2024). Additionally, high-level concepts such as sentiment are encoded in linear activation structures (Tigges et al., 2023), highlighting the model's ability to organize information hierarchically.

Another line of research suggests that certain Transformer layers contribute little to the model's final prediction. By analyzing how probability distributions evolve across blocks, researchers observed that in many cases, a model's prediction stabilizes early—once a token becomes the most probable, it remains unchanged until the final layer. These stabilization points, known as saturation events, suggest that the model's later layers primarily refine rather than reshape its output (Geva et al., 2022). Further studies confirmed that even lower-ranked tokens follow the same pattern once the top-1 prediction stabilizes (Lioubashevski et al., 2025). Moreover, experimental evidence shows that middle blocks can be removed or swapped with minimal impact on performance (Sun et al., 2025).

These findings have led to the prevailing belief that some Transformer blocks are inherently unimportant. In this work, we revisited this assumption by showing that a block's relevance can vary significantly depending on the task—suggesting that global conclusions about importance may overlook task-specific dynamics.

## A.2 MEASURING BLOCK RELEVANCE

Most research on measuring block relevance has been conducted in the context of structured pruning (Men et al., 2025; Gromov et al., 2025; He et al., 2024; Kim et al., 2024; Ma et al., 2023; Zhang et al., 2024b; Yang et al., 2026; Sieberling et al., 2024). The goal is to remove the least relevant blocks while preserving model performance, which has led to the development of several techniques for estimating a block's importance.

A foundational but now outdated approach is magnitude-based pruning, which removes blocks based on their parameter magnitudes. While widely used for individual weight pruning (Li et al., 2016), this method proved too simplistic at the block level. Still, it served as a useful baseline in the early development of structured pruning techniques.

More recent work has focused on proxy-based relevance scores that analyze how much a block transforms its input. One popular class of methods uses cosine similarity between a block's input and output, assuming that low transformation implies low relevance (Men et al., 2025; Gromov et al., 2025; He et al., 2024). Other studies rely on Taylor expansion techniques to estimate the change in loss when a weight or block is removed, providing a more gradient-informed view of importance (Kim et al., 2024; Ma et al., 2023).

Another set of methods evaluates relevance by comparing the pruned model's output to the original model's, using metrics like cosine similarity, norm differences, and divergence-based measures. Zhang et al. (2024b), for example, employ Jensen-Shannon divergence to guide pruning and achieve state-of-the-art results. Follow-up work builds on this idea using KL divergence, a closely related metric: Yang et al. (2026) apply it as part of a multi-step strategy to create smaller models tailored to code generation, while Sieberling et al. (2024) combine it with a novel selection algorithm that prunes blocks jointly rather than iteratively.

Perplexity-based metrics are also used, especially in language modeling, where blocks are considered irrelevant if their removal does not significantly increase perplexity (Kim et al., 2024; Zhong et al., 2025; Song et al., 2024). Beyond pruning-specific methods, some studies draw on game-theoretic tools, such as approximations of Shapley values, to assess a block's contribution to the model's output in a more theoretically grounded way (Zhang et al., 2024a; Siddiqui et al., 2024).

While most pruning research focuses on overall effectiveness, few works ask whether a block's relevance remains stable as the model is progressively pruned. He et al. (2024) and Lu et al. (2024), for instance, compare one-shot pruning—where relevance scores are computed a single time and used to select all blocks to prune—with iterative pruning, where relevance is recalculated and re-ranked after each pruning step. Their findings suggest that one-shot pruning can match or even outperform iterative pruning for structured sparsity. However, their analyses center on end-task accuracy rather than how relevance itself shifts during the process. This leaves an important question unanswered: Does pruning change block relevance? A question that we answered in Section C.5.

While most methods rely on a single calibration dataset to assess relevance, some recent studies have started exploring the generalizability of relevance scores across datasets. He et al. (2024) found that relevance maps computed via cosine similarity appear largely consistent across datasets, leading them to conclude that certain layers may be universally important or unimportant. This perceived dataset-agnostic behavior motivated their decision to use a single calibration dataset throughout their experiments. Their findings also connect to saturation-based analyses (Geva et al., 2022; Lioubashevski et al., 2025), which similarly suggest that once a model's prediction stabilizes, later computations may be less critical.

We took He et al. (2024) as our primary baseline because they provide one of the few systematic attempts to visualize and quantify block relevance across tasks. Their heatmaps offered a clear point of comparison for our own cross-task analysis, which was built on their setup but replaces similarity-based relevance with a task-grounded, accuracy-based metric.

## B    PROOF OF THEOREM 1

### B.1    AUXILIARY RESULT

Before proving Theorem 1, let's first prove the following theorem:

**Theorem 2** *For any $\epsilon > 0$ and unlabeled calibration dataset $\mathbb{D} = \{s^{(i)}\}_{i=1}^{N}$, there exists a decoder-only Transformer $f^L$ with $L \geq 3$ and a labeling function $\mathcal{L} : \mathbb{D} \rightarrow \{0, 1\}$ satisfying the following conditions:*

1. *There exists an intermediate layer $l \in \{1, \ldots, L - 2\}$ such that $\mathrm{CosSimScore}(l; \mathbb{D}) = \epsilon$, and $\mathrm{CosSimScore}(i; \mathbb{D}) > \epsilon$ for all $i \neq l$.*

2. *The full model achieves perfect accuracy: $f^L(s^{(i)}) = \mathcal{L}(s^{(i)})$ for all $s^{(i)} \in \mathbb{D}$, but removing layer $l$ causes the model's accuracy to drop to zero: $f^L_{-l}(s^{(i)}) \neq \mathcal{L}(s^{(i)})$ for all $s^{(i)} \in \mathbb{D}$.*

This result can be viewed as a simplified version of Theorem 1, where the labeling function is binary and freely chosen, rather than being fixed by the dataset $\mathbb{D}$.

Let $E(s^{(i)}) = \boldsymbol{X}^{(0,i)}$ denote the embedding of a sequence $s^{(i)}$, where $\boldsymbol{X}^{(l,i)} \in \mathbb{R}^{n \times d}$, with $n$ the number of tokens and $d$ the hidden dimension. The transformation at block $l$ is given by

$$\boldsymbol{X}^{(l+1,i)} = \boldsymbol{X}^{(l,i)} + f\big(\boldsymbol{X}^{(l,i)}; \boldsymbol{\theta}^{(l)}\big).$$

A decoder-only transformer with $L$ blocks is then

$$f^L(s^{(i)}) = U\left( E(s^{(i)}) + \sum_{l=0}^{L-1} f\big(\mathbf{X}^{(l,i)}; \boldsymbol{\theta}^{(l)}\big) \right),$$

where $U(\cdot)$ denotes the final transformation applied to the output of the last block (e.g., an unembedding layer for next-token prediction or a classification head).

We also define the model obtained by removing block $l$, denoted $f_{-l}^L$. In this case, the hidden state $X^{(l-1)}$ is directly connected to block $l + 1$, bypassing block $l$. Formally,

$$f_{-l}^L(s^{(i)}) = U\left( E(s^{(i)}) + \sum_{\substack{k=0 \\ k \neq l}}^{L-1} f\big(\boldsymbol{X}^{(k,i)}; \boldsymbol{\theta}^{(k)}\big) \right),$$

with the convention that

$$\boldsymbol{X}^{(l+1,i)} = \boldsymbol{X}^{(l-1,i)} + f\big(\boldsymbol{X}^{(l-1,i)}; \boldsymbol{\theta}^{(l-1)}\big) \quad \text{for the pruned model.}$$

Let $\mathbf{1}_n$ denote the column vector of size $n$ with all entries equal to one, and $\mathbf{0}_n$ the zero vector of the same size.

Consider the embedding function

$$E(s_i) = \begin{bmatrix} \mathbf{0}_{n^{(i)}} & \delta \cdot \mathbf{1}_{n^{(i)}} & \mathbf{0}_{n^{(i)}} \end{bmatrix}, \qquad \forall s^{(i)} \in \mathbb{D},$$

with hidden dimension $d = 3$ and $\delta > 0$. Thus, every token in the vocabulary has the same embedding.

Define the labeling function $\mathcal{L}(s^{(i)}) = 0$ for all $s^{(i)} \in \mathbb{D}$, i.e., all sentences belong to the same class. The final transformation is a standard classification head

$$U(\boldsymbol{X}) = \underset{j \in \{0,1\}}{\arg\max} \operatorname{softmax}\big[\boldsymbol{X}_{n^{(i)}} \boldsymbol{W}^U\big]_j,$$

where $\boldsymbol{X}_{n^{(i)}}$ is the representation of the last token, and

$$\boldsymbol{W}^U = \begin{bmatrix} 1 & 0 \\ 0 & 1 \\ 0 & 0 \end{bmatrix}.$$

We now construct three blocks as follows (with $M \gg 1$):

$$\boldsymbol{X}^{(1,i)} = \boldsymbol{X}^{(0,i)} + \begin{bmatrix} \mathbf{0}_{n^{(i)}} & \mathbf{0}_{n^{(i)}} & M \cdot \mathbf{1}_{n^{(i)}} \end{bmatrix},$$

$$\boldsymbol{X}^{(2,i)} = \boldsymbol{X}^{(1,i)} + \begin{bmatrix} \delta \cdot \mathbf{1}_{n^{(i)}} & \mathbf{0}_{n^{(i)}} & \mathbf{0}_{n^{(i)}} \end{bmatrix},$$

$$\boldsymbol{X}^{(3,i)} = \boldsymbol{X}^{(2,i)} + \begin{bmatrix} \delta M \cdot \mathbf{1}_{n^{(i)}} & \mathbf{0}_{n^{(i)}} & -M \cdot \mathbf{1}_{n^{(i)}} \end{bmatrix}.$$

Each Transformer block contains a feed-forward network of the form

$$\text{FFN}(\boldsymbol{X}) = \text{ReLU}(\boldsymbol{X}\boldsymbol{W}_1 + \mathbf{1}_n \boldsymbol{b}_1^\top)\boldsymbol{W}_2 + \mathbf{1}_n \boldsymbol{b}_2^\top,$$

where $\boldsymbol{W}_1, \boldsymbol{W}_2 \in \mathbb{R}^{d \times d}$ and $\boldsymbol{b}_1, \boldsymbol{b}_2 \in \mathbb{R}^d$. Note that the bias vectors are written as $\mathbf{1}_n \boldsymbol{b}^\top$ so that dimensions match for sequence length $n$.

To enforce that the multi-head attention does not modify the representation, we set its output to zero, so that the residual connection yields the identity mapping.

For the FFN, we choose $\boldsymbol{W}_1 = \boldsymbol{I}$, $\boldsymbol{W}_2 = \boldsymbol{I}$, and $\boldsymbol{b}_1 = \boldsymbol{0}$, so the residual effect comes only from $\boldsymbol{b}_2$. Specifically:

- In Block 1, set $\boldsymbol{b}_2 = (0, 0, M)^\top$ to add $M$ in the third coordinate.
- In Block 2, set $\boldsymbol{b}_2 = (\delta, 0, 0)^\top$ to add $\delta$ in the first coordinate.

- In Block 3, we instead choose

$$\boldsymbol{W}_2 = \begin{bmatrix} M & 0 & 0 \\ 0 & 0 & 0 \\ 0 & 0 & 0 \end{bmatrix}, \quad \boldsymbol{b}_2 = (0, 0, -M)^\top,$$

so that the FFN contributes the transformation

$$\begin{bmatrix} \delta M \cdot \mathbf{1}_{n^{(i)}} & \mathbf{0}_{n^{(i)}} & -M \cdot \mathbf{1}_{n^{(i)}} \end{bmatrix}.$$

With this construction, the model output is

$$f^3(s^{(i)}) = U([\delta(M+1) \cdot \mathbf{1}_{n^{(i)}} \quad \delta \cdot \mathbf{1}_{n^{(i)}} \quad \mathbf{0}_{n^{(i)}}]) = 0 = \mathcal{L}(s_i),$$

while pruning the second block yields

$$f^3_{-1}(s^{(i)}) = U([\mathbf{0}_{n^{(i)}} \quad \delta \cdot \mathbf{1}_{n^{(i)}} \quad \mathbf{0}_{n^{(i)}}]) = 1 \neq \mathcal{L}(s_i).$$

Finally, we compute the cosine-similarity scores:

$$\mathrm{CosSimScore}(0; \mathbb{D}) = 1 - \frac{\delta}{\sqrt{\delta^2 + M^2}},$$

$$\mathrm{CosSimScore}(1; \mathbb{D}) = 1 - \frac{\sqrt{\delta^2 + M^2}}{\sqrt{2\delta^2 + M^2}},$$

$$\mathrm{CosSimScore}(2; \mathbb{D}) = 1 - \frac{\delta(M+1)}{\sqrt{2\delta^2 + M^2}\sqrt{1 + M^2}}.$$

As $M \to \infty$, we obtain

$$\mathrm{CosSimScore}(0; \mathbb{D}) \to 1, \quad \mathrm{CosSimScore}(1; \mathbb{D}) \to 0, \quad \mathrm{CosSimScore}(2; \mathbb{D}) \to 1.$$

Thus, by choosing appropriate values of $M$ and $\delta$, we can ensure

$$\mathrm{CosSimScore}(1; \mathbb{D}) = \epsilon,$$

which completes the proof for Theorem 2.

It is also worth noting that this argument can be extended to multiple dimensions that do not affect the task, rather than relying on a single one. In this way, instead of requiring a large value of $M$, we can use several smaller dimensions $M_1, M_2, ..., M_d$.

Finally, one might worry that this construction would fail in practice because each block also includes a LayerNorm operation applied after the residual aggregation. However, in our setup every row of $\boldsymbol{X}^{(l)}$ is identical, so each token representation has the same mean and variance at every step. Consequently, the effect of LayerNorm is deterministic and can be exactly canceled out by choosing the LayerNorm parameters $(\gamma, \beta)$ appropriately. In particular, setting $\gamma$ and $\beta$ to rescale and shift the normalized vectors recovers the pre-normalized representation, ensuring that LayerNorm does not alter the intended behavior of the construction.

## B.2 GENERAL CASE

We first show that a decoder-only Transformer can trivially overfit any labeled calibration dataset $\mathbb{D} = \{(s^{(i)}, y^{(i)})\}_{i=1}^N$, where $s^{(i)} \neq s^{(j)}$ for $i \neq j$ and $y^{(i)} \in \{0, \ldots, C-1\}$.

Suppose that the tokenizer assigns one token to each sequence $s^{(i)}$. Define an embedding function $E(\cdot)$ such that

$$E(s^{(i)}) = \boldsymbol{X}^{(i)} \in \mathbb{R}^{1 \times C}.$$

If we let

$$E(s^{(i)}) = (\boldsymbol{e}^{(y^{(i)}+1)})^\top,$$

where $\boldsymbol{e}^{(j)}$ is the $j$-th standard basis vector in $\mathbb{R}^C$, then the classification head

$$U(\boldsymbol{X}) = \underset{j \in \{0, \ldots, C-1\}}{\arg\max} \, \mathrm{softmax}\big[\boldsymbol{X}\boldsymbol{W}^U\big]_j,$$

with

$$\boldsymbol{W}^U = \begin{bmatrix} (\boldsymbol{e}^{(1)})^\top \\ \vdots \\ (\boldsymbol{e}^{(C)})^\top \end{bmatrix},$$

perfectly classifies the dataset, i.e. $f(s^{(i)}) = y^{(i)}$. Thus, the model can memorize the dataset without any Transformer blocks, using only embeddings and the unembedding matrix.

We now extend this idea to construct a model satisfying the conditions of Theorem 1. Let the hidden dimension be $d = 2C + 1$. Define the embedding as

$$E(s^{(i)}) = \delta \cdot (\boldsymbol{e}^{(C+y^{(i)}+2)})^\top \in \mathbb{R}^d,$$

so that each input is mapped into a unique coordinate among the last $C$ dimensions (beyond the first $C + 1$).

We construct three Transformer blocks as follows (with $M \gg 1$):

- **Block 1.** Adds $M$ to coordinate $C + 1$:

$$\boldsymbol{X}^{(1,i)} = \boldsymbol{X}^{(0,i)} + M \cdot \boldsymbol{e}^{(C+1)}.$$

- **Block 2.** Adds $\delta \cdot \boldsymbol{e}^{(y^{(i)}+1)}$, i.e. a one-hot signal in the first $C$ coordinates corresponding to the correct class:

$$\boldsymbol{X}^{(2,i)} = \boldsymbol{X}^{(1,i)} + \delta \cdot \boldsymbol{e}^{(y^{(i)}+1)}.$$

- **Block 3.** Amplifies the signal in the first $C$ coordinates by $(M - \delta)$, subtracts $M$ from coordinate $C + 1$, and adds a misleading one-hot vector from the last $C$ dimensions:

$$\boldsymbol{X}^{(3,i)} = \boldsymbol{X}^{(2,i)} + (M - \delta) \cdot \boldsymbol{e}^{(y^{(i)}+1)} - M \cdot \boldsymbol{e}^{(C+1)} + \delta \cdot \boldsymbol{e}^{(C-y^{(i)})}.$$

As in the proof of Theorem 2, we ensure that multi-head attention acts as the identity by setting its output projection $\boldsymbol{W}_O = 0$, and we use the feed-forward networks with suitable $(\boldsymbol{W}_1, \boldsymbol{W}_2, \boldsymbol{b}_1, \boldsymbol{b}_2)$ to realize the desired additive transformations.

After the three blocks, the first $C$ coordinates of $\boldsymbol{X}^{(3,i)}$ are dominated by $(M - \delta + \delta) \cdot \boldsymbol{e}^{(y^{(i)}+1)} = M \cdot \boldsymbol{e}^{(y^{(i)}+1)}$, while the misleading additions are suppressed. Thus, the classifier $U$ correctly outputs $y^{(i)}$ for all $i$, and the model achieves perfect accuracy.

However, if Block 2 is removed, then the model never inserts the signal in the first $C$ coordinates. Block 3 then only contributes spurious information, and the classification head produces incorrect labels for all samples. Therefore, the pruned model fails completely.

Finally, as in Theorem 2, we compute the cosine similarity scores for each block. By taking $M \to \infty$ and choosing $\delta$ appropriately, we ensure that Block 2 attains $\mathrm{CosSimScore}(l; \mathbb{D}) = \epsilon$, while the others approach 1. Thus, the theorem follows.

Two final remarks are worth noting. First, if the number of classes $C$ is odd, the pruned model may not achieve zero accuracy. Specifically, instances assigned to class $\frac{C-1}{2}$ will be classified correctly, as the misleading signal coincides with the correct label. This issue is trivial to resolve by adjusting the label assignment or class structure.

Second, as in the proof of Theorem 2, the presence of normalization layers does not invalidate the construction. This is because the mean and variance of each token representation within a block remain constant across instances, ensuring that normalization does not interfere with the mechanism underlying the proof.

## C   FURTHER ANALYSIS ABOUT RELEVANCE CONSISTENCY ACROSS DATASETS

In this section we go deeper in the analysis done in Section 5.1, about the work from He et al. (2024).

## C.1 Implementation Details

All experiments were conducted on pre-trained models, using code based on the EleutherAI LM Evaluation Harness (Gao et al., 2024) for our accuracy-based scores. We used a batch size of 4 and ran evaluations on NVIDIA RTX A6000 and RTX 4090 GPUs.

To compute cosine similarity relevance scores, we used the same hardware and followed the methodology introduced in Section 3, based on the implementation from He et al. (2024). Each sample in this method is a full input sequence matching the model's context length (e.g., 4096 tokens for Mistral-7B), constructed by concatenating multiple dataset instances until the required token length is reached. Because instance lengths vary across datasets, the number of instances per sample also varies. Following He et al. (2024), we use 256 such samples per dataset for C4, LIMA, CodeAlpaca, and MathInstruct.

For fairness, we used the same dataset instances to compute our accuracy-based relevance scores. However, unlike cosine similarity, we did not concatenate instances into long sequences. Instead, we evaluated next-token prediction at the instance level, computing accuracy on the last token of each instance. Thus, while the underlying data is shared, the two metrics differ in their evaluation granularity.

For the remaining datasets, we used the training split associated with each task and modified the input format used during relevance scoring to compute cosine similarity scores. Specifically, instead of generating full answer phrases, we presented all answer options explicitly (e.g., "A", "B", "C", "D") within the prompt and computed the probability of generating only the correct option token. This adjustment was necessary to ensure the model received all relevant information required for task evaluation. In contrast, no such modification was needed for our accuracy-based metric, as we followed standard LM evaluation protocols for multiple-choice tasks.

Following these protocols (Gao et al., 2024), we constructed input prompts by concatenating the context, question, and each answer option individually. For each example, we computed the total log-probability of the full prompt associated with each option and selected the one with the highest value. We report normalized accuracy, which adjusts log-probabilities for option length to ensure fairness between longer and shorter candidates. A prediction is counted as correct if the selected option matches the gold label.

## C.2 Normalization Details

To complement our block relevance visualizations and quantify how our accuracy-based score captures more variation across datasets than cosine similarity, we compute the variance in relevance across tasks for both methods. For each model and method, we first apply z-score normalization to the block relevance scores, then calculate the variance across datasets for each of the 32 layers—yielding 32 variance values per model-method combination. In Figure 5, we report the mean variance and standard deviation error bars for each model and method.

## C.3 Wilcoxon signed-rank test

When comparing our accuracy-based relevance to cosine similarity, we found significantly higher variance using our metric. In fact, the Wilcoxon signed-rank test (Wilcoxon, 1992) resulted in the following values: p-value = 1.7e-7, W-value = 20 for Mistral; p-value = 8.8e-9, W-value = 7 for OLMo; and p-value = 4.6e-10, W-value = 0 for Pythia, respectively.

## C.4 Relevance During Training

Block relevance patterns evolve during training, but in markedly different ways depending on the metric. Our accuracy-based metric (Figures 7A and 7B) displays a chaotic behavior through training, with some blocks gaining or losing relevance between checkpoints without following smooth trends. While certain blocks in OLMo tend to increase in relevance, these changes are rarely monotonic. This fluctuation suggests that blocks may take on transient, adaptive roles throughout training—dynamics that cosine similarity tends to obscure.

Cosine similarity (Figures 7C and 7D) reveals consistent patterns across models. For both models, most blocks either maintain their relevance scores or gradually increase throughout training. This suggests that some blocks increasingly modify their inputs as training progresses. However, it's important to note that this does not necessarily reflect how much each block contributes to the model's output.

Figures 8, 9 and 10 present the results on CodeAlpaca, C4 and LIMA, respectively, using OLMo. As with MathInstruct, the cosine similarity-based relevance (bottom figures) produces nearly identical heatmaps across datasets, reinforcing the metric's stability and dataset-agnostic nature. Interestingly, we also find a pattern not discussed in prior work—block 2 shows a non-monotonic trajectory where its relevance increases at early stages and later decreases, a pattern that could be studied in future works.

In contrast, the accuracy-based relevance (top figures) continues to show less consistent and less interpretable patterns. While some blocks exhibit periods of increased or decreased relevance, there are no clear, sustained trends comparable to those seen with cosine similarity.

Figures 11 to 13 show the results for the same experiment, but with Pythia on the same four datasets used in previous sections. Unlike the OLMo figures, we apply a separate color scale for block 1 in the cosine similarity plots (bottom figures) for all Pythia figures. This is necessary because the relevance values of the first block are significantly higher than the rest—using a single color scale would make differences between blocks 2 to 31 nearly invisible.

As with OLMo, cosine similarity yields nearly identical relevance patterns across datasets, reinforcing the observation that this metric is largely insensitive to the specific task. However, a new behavior emerges in Pythia: some blocks show an initial drop in relevance between the first and second checkpoints, but then stabilize or fluctuate rather than continue decreasing. This, along with the unusual pattern in block 2 in OLMo, suggests that certain relevance dynamics may be model-specific. More precisely, we suspect they may be seed-specific: different initializations of the same model trained on the same data could produce distinct relevance trajectories.

In contrast, our accuracy-based metric continues to show no clear, smooth patterns across training steps, and exhibits noticeable differences between datasets. One particularly interesting finding is that blocks 17 to 31 appear nearly irrelevant under cosine similarity for all datasets—yet our method shows that pruning some of these blocks can significantly hurt performance. This further illustrates that cosine similarity can miss important functional contributions of blocks, reinforcing the need for task-aware relevance measures.

Finally, through our experiments, we do not observe a clear relationship between block relevance patterns and the model's accuracy gains throughout training for either metric. In other words, changes in block relevance do not directly correlate with improvements in overall performance, highlighting the complexity of the internal dynamics involved during model learning.

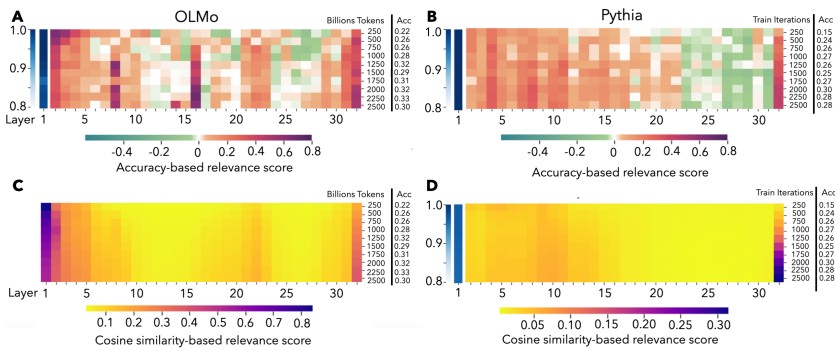

Figure 7: Block relevance during training in OLMo (left) and Pythia (right) on the MathInstruct dataset. **A**, each row corresponds to a model checkpoint trained on a given number of tokens in billions (OLMo) or train iterations in millions (Pythia on **B**), with accuracy reported on the y-axis. **A**, **B**, Accuracy-based score. **C**, **D**, Cosine-similarity score.

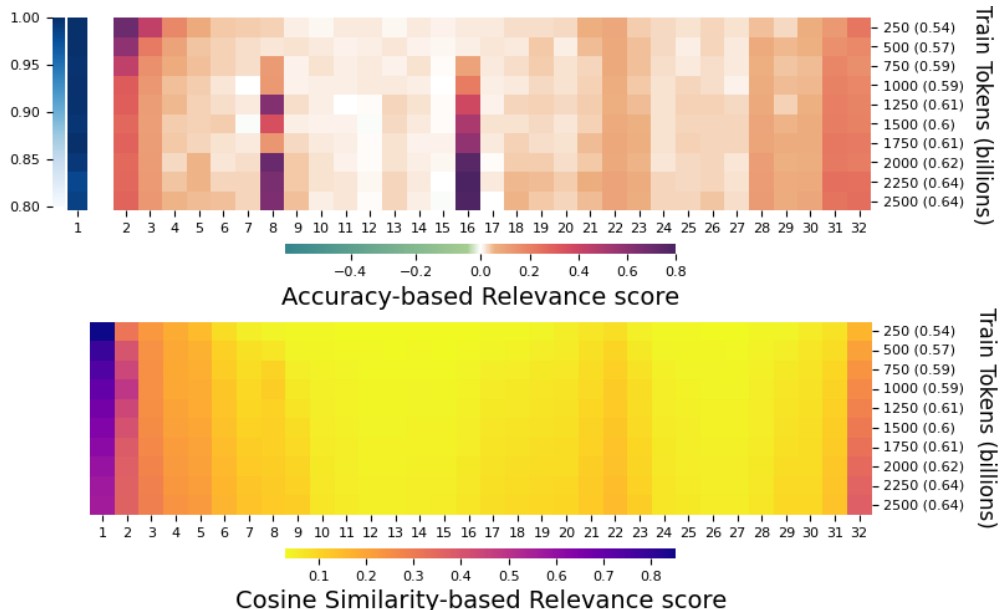

Figure 8: Block relevance during training in OLMo on the CodeAlpaca dataset. Each row corresponds to a model checkpoint trained on a given number of tokens in billions (shown on the y-axis), with accuracy reported in parentheses. (Top) Accuracy-based score. (Bottom) Cosine-similarity score.

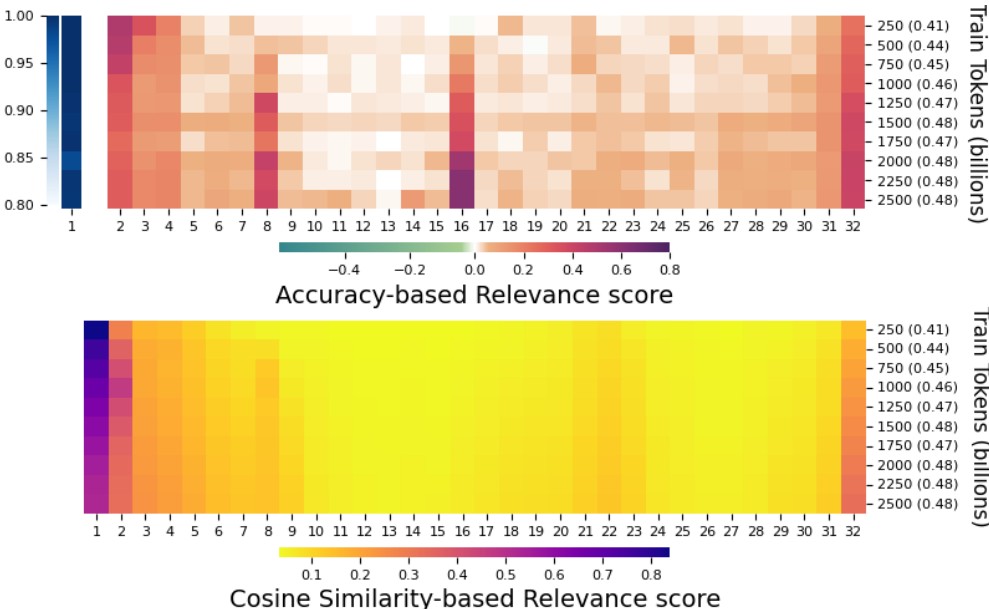

Figure 9: Block relevance during training in OLMo on the C4 dataset. Each row corresponds to a model checkpoint trained on a given number of tokens in billions (shown on the y-axis), with accuracy reported in parentheses. (Top) Accuracy-based score. (Bottom) Cosine-similarity score.

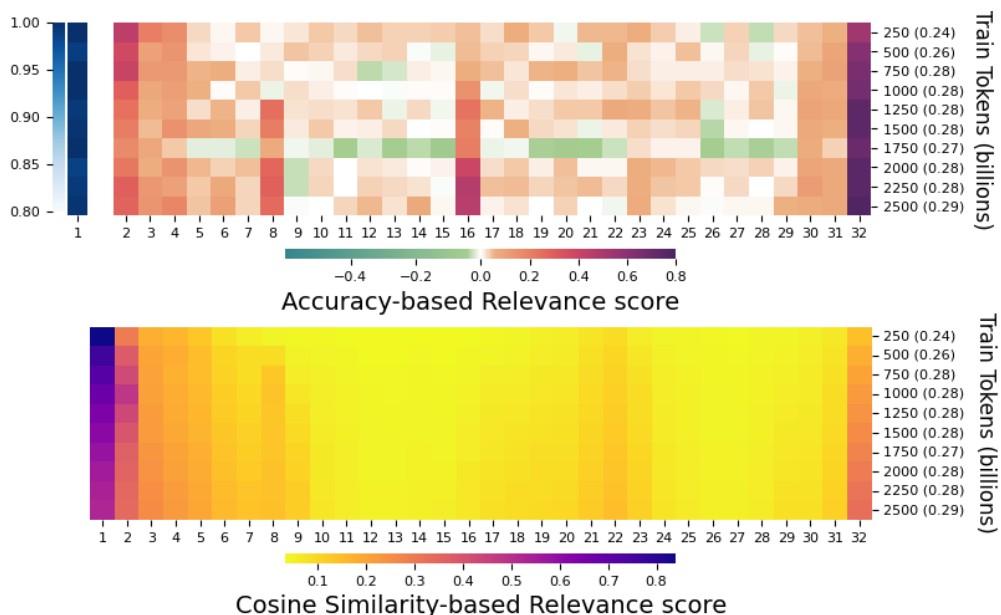

Figure 10: Block relevance during training in OLMo on the LIMA dataset. Each row corresponds to a model checkpoint trained on a given number of tokens in billions (shown on the y-axis), with accuracy reported in parentheses. (Top) Accuracy-based score. (Bottom) Cosine-similarity score.

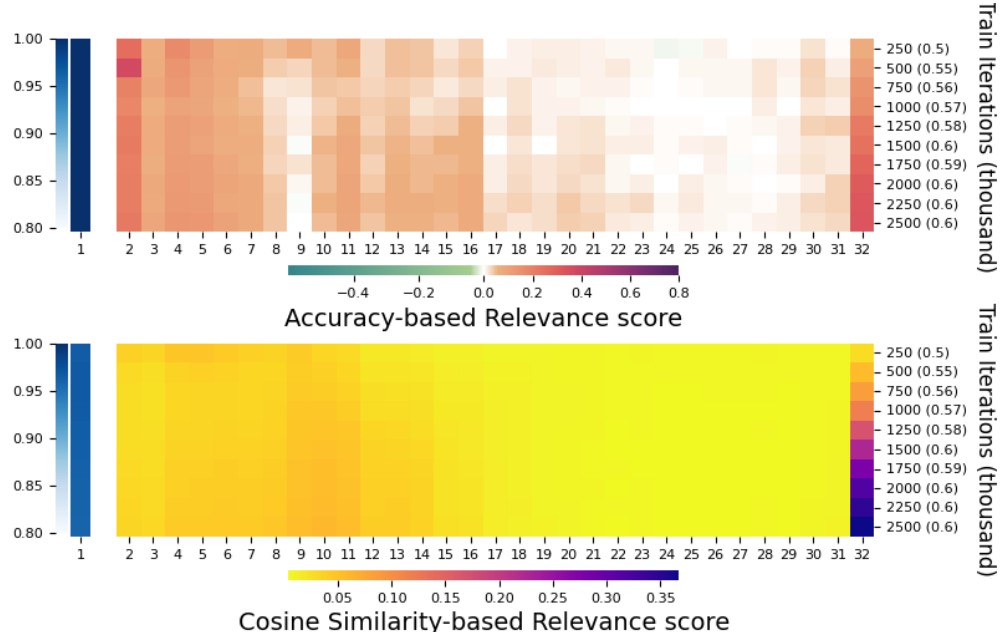

Figure 11: Block relevance during training in Pythia on the CodeAlpaca dataset. Each row corresponds to a model checkpoint trained with a given number of iterations in thousand (shown on the y-axis), with accuracy reported in parentheses. (Top) Accuracy-based score. (Bottom) Cosine-similarity score.

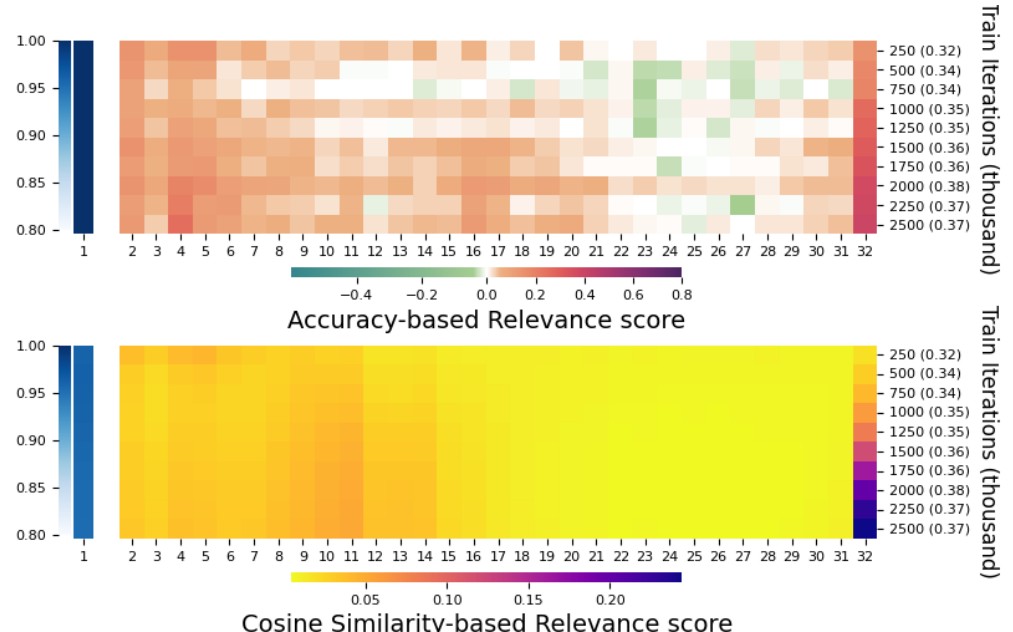

Figure 12: Block relevance during training in Pythia on the C4 dataset. Each row corresponds to a model checkpoint trained with a given number of iterations in thousand (shown on the y-axis), with accuracy reported in parentheses. (Top) Accuracy-based score. (Bottom) Cosine-similarity score.

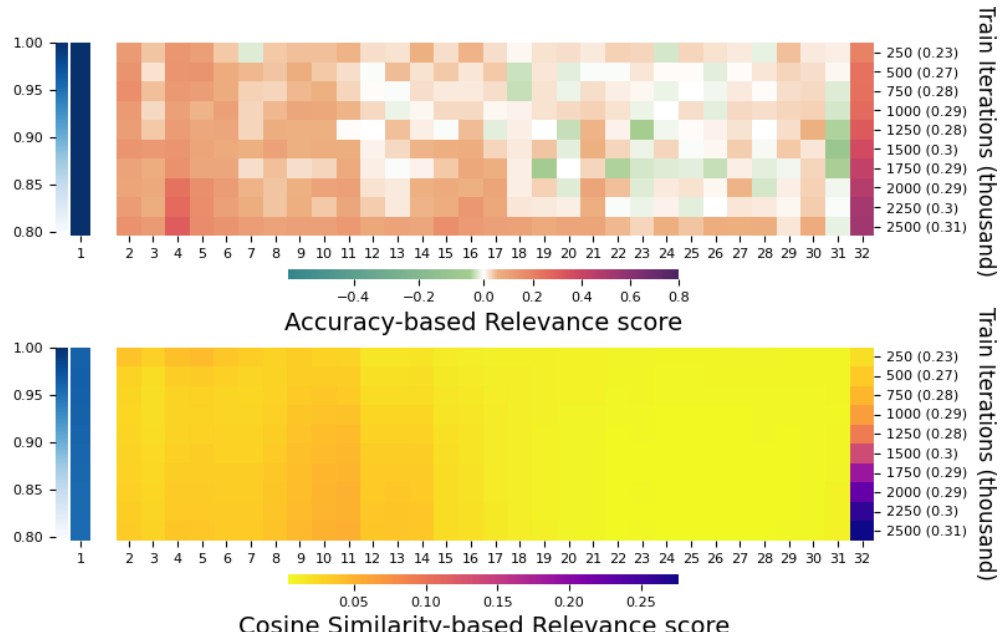

Figure 13: Block relevance during training in Pythia on the LIMA dataset. Each row corresponds to a model checkpoint trained with a given number of iterations in thousand (shown on the y-axis), with accuracy reported in parentheses. (Top) Accuracy-based score. (Bottom) Cosine-similarity score.

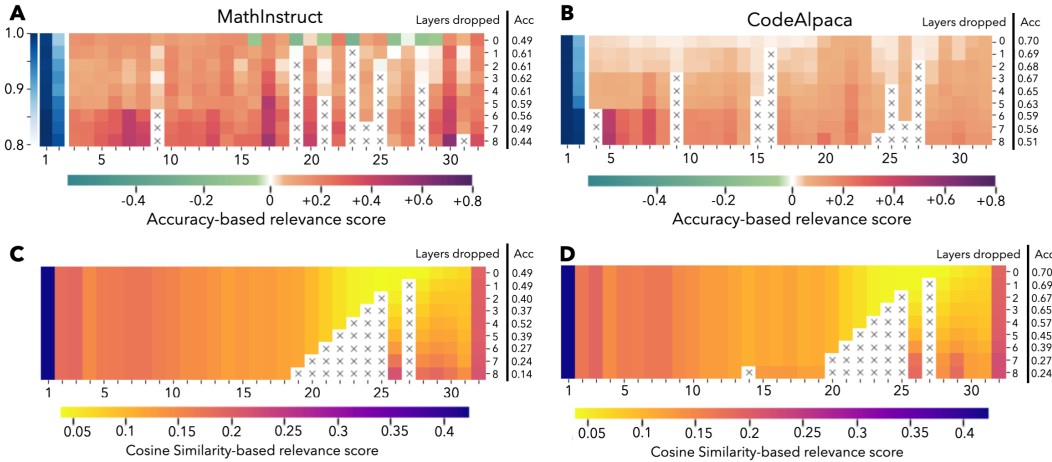

Figure 14: Block relevance on Mistral in MathInstruct (left) and CodeAlpaca (right) as blocks are iteratively pruned. **A**, at each row the least relevant block, according to the Accuracy-based score of Mistral on MathInstruct, is removed and shown with a gray cross. The accuracy of the pruned model is shown on the right. **B**, the same on CodeAlpaca. **C**, **D**, using cosine-similarity score.

## C.5    Relevance During Pruning

Pruning significantly changes block relevance—especially under our accuracy-based metric. As model blocks are pruned, we observe that certain blocks increase in importance while others become less critical. These shifts reveal that accuracy-based relevance captures latent dependencies and compensatory dynamics between layers. To better understand how these shifts in relevance emerge, we performed iterative structured pruning on Mistral-7B. At each step, we (1) compute block relevance using either our accuracy-based method or cosine similarity, (2) remove the least relevant block, and (3) repeat steps one and two until 25% of blocks are pruned. Figure 14 shows results for MathInstruct and CodeAlpaca, while Figure 15 and Figure 16 show results for C4 and LIMA respectively. The figures also report the accuracy for the same dataset used for pruning.

Pruning using the accuracy-based score (Figures 14A and 14B) reveals complex dynamics. First, after pruning a block, earlier (closer to the input) and/or later (closer to the output) blocks can gain relevance. For example, in MathInstruct (Figure 14A), block 17—initially of moderate importance—becomes highly relevant once later blocks are removed, suggesting that pruning can reassign or expose latent functional roles. Second, blocks with negative relevance (green blocks) become neutral or positive after pruning. For example, in the first row of Figure 14A, several green blocks change behavior after pruning block 23, implying that they were not inherently harmful but instead interacted negatively with it. Third, blocks with high relevance decreased their value after pruning. For example, we observe that block 31 becomes less relevant after block 23 is removed, which we speculate reflects a compensatory role—block 31 may have been mitigating the detrimental effects of block 23, a pattern aligning with prior findings on corrective behavior (Geva et al., 2022). These examples suggest that our metric can be a tool to study the inner workings of transformers.

As expected, when pruning using our method, we observed differences in relevance at the dataset level. In MathInstruct, pruning blocks triggers sharp shifts in relevance, while in CodeAlpaca, the relevance landscape remains relatively stable in the early pruning steps. Notably, CodeAlpaca lacks negatively relevant blocks at initialization, suggesting less redundancy or a more uniform functional distribution, among other possible explanations. This phenomenon opens new avenues for research.

On the other hand, under cosine similarity (Figures 14C and  14D), we observe that relevance changes after pruning are generally local and limited. Only the later blocks of the network, those positioned after the pruned block, display relevance changes according to this measure. For instance, in MathInstruct, pruning block 27 results in slight increases of cosine-similarity score in later blocks, while earlier blocks remain unaffected. Even though one might expect this behavior given the local nature of the metric, this explanation is only partially correct. Since cosine similarity is computed locally, only blocks following the pruned one can exhibit changes in relevance. Mathematically, these changes could be either increases or decreases; however, in practice, we observe only increases.

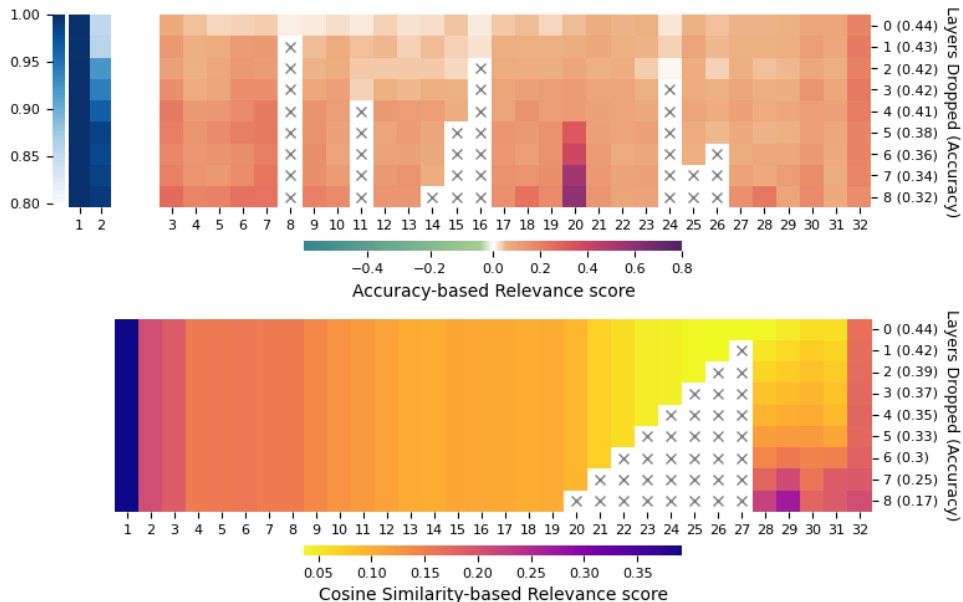

Figure 15: Block relevance in Mistral on the C4 dataset as layers are incrementally pruned. In each row, the least relevant block (according to the corresponding metric) is removed and shown with a gray cross. The accuracy of the pruned model is shown in parentheses. (Top) Accuracy-based score. (Bottom) Cosine-similarity score.

When using accuracy-based relevance, iterative pruning produces a different model compared to one-shot pruning, which removes all least-relevant blocks simultaneously based on initial relevance scores. As shown in Figure 14, our metric reveals that block relevance changes significantly after each pruning step, with new dependencies and compensatory patterns emerging across layers. For example, under one-shot pruning, blocks 16, 19, 21, 23, 26, 27, 28, and 29 would be removed from Mistral (Figure 14A first row); in this case, the pruned model would exhibit an accuracy of 0.22 (data not shown). In contrast, based on iterative pruning, we removed different blocks, resulting in a pruned model accuracy of 0.44. Our results indicate that one-shot pruning may not be suitable when employing accuracy-based relevance. In contrast, cosine similarity yields nearly identical results for both one-shot and iterative pruning since relevance scores remain largely stable throughout the pruning steps.

Regarding C4 and LIMA datasets. We observe similar patterns to those previously discussed: our accuracy-based relevance scores reveal richer dynamics than cosine similarity.

With the accuracy-based metric, we see both increases and decreases in relevance as pruning progresses. In rare cases, such as block 1, relevance remains stable throughout. In contrast, cosine similarity mostly shows increasing relevance in blocks that follow the pruned one, while other blocks remain largely unaffected.

An interesting pattern emerges in both C4 and LIMA: block 20 consistently increases in relevance under our metric. This may suggest a shared functional role between these two tasks, though it may also be coincidental. A deeper investigation into this connection would be valuable.

Regarding the comparison between one-shot and iterative pruning, we noted that the two approaches often select different sets of blocks for removal. However, the reasons for these divergences differ depending on the relevance metric.

For cosine similarity (Figures 14C and 14D), the evolution is mostly predictable. As discussed earlier, pruning a block tends to increase the similarity scores of subsequent blocks. As a result, iterative pruning diverges from one-shot pruning primarily when the least relevant block is not one of the later-positioned layers. For example, in MathInstruct, block 28 initially had low relevance, but pruning earlier blocks (e.g., block 27) increased its relevance, causing it to be excluded from later pruning steps. A similar shift happens with block 26. If the initial relevance ordering of blocks

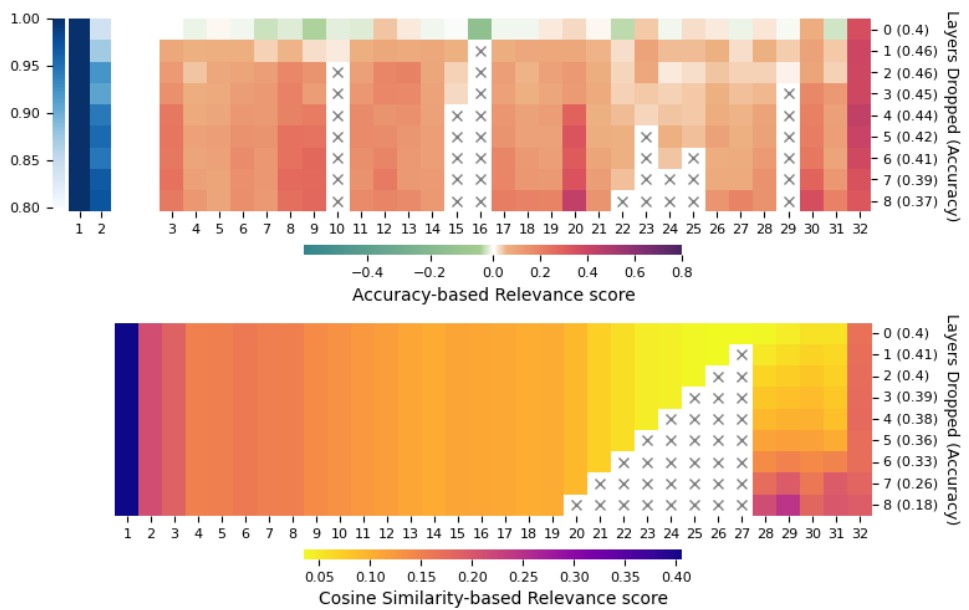

Figure 16: Block relevance in Mistral on the LIMA dataset as layers are incrementally pruned. In each row, the least relevant block (according to the corresponding metric) is removed and shown with a gray cross. The accuracy of the pruned model is shown in parentheses. (Top) Accuracy-based score. (Bottom) Cosine-similarity score.

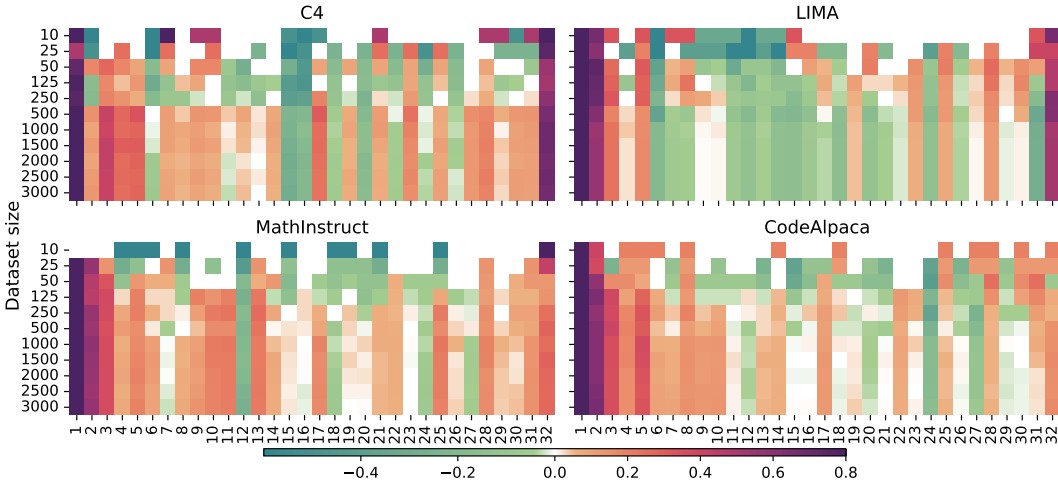

Figure 17: Block relevance in LLaMA-3-8B across 4 datastes. In each row, we used a different size of the dataset to compute the metric.

21–28 had been strictly decreasing, both pruning methods would have selected the same blocks. The observed deviations result from small, local shifts in relevance caused by positional effects during pruning.

## C.6    SIZE OF THE CALIBRATION DATASET

To further analyze our metric, we applied it to LLaMA-3-8B using four different datasets: C4, LIMA, MathInstruct, and CodeAlpaca, each with varying dataset sizes. Figure 17 presents the results of this experiment. We observe that after using approximately 500 samples, the heatmaps begin to converge toward the scores computed with 3,000 samples. While some differences remain noticeable, they are not critical for the overall ranking; in other words, the sets of relevant and irrelevant blocks remain consistent.

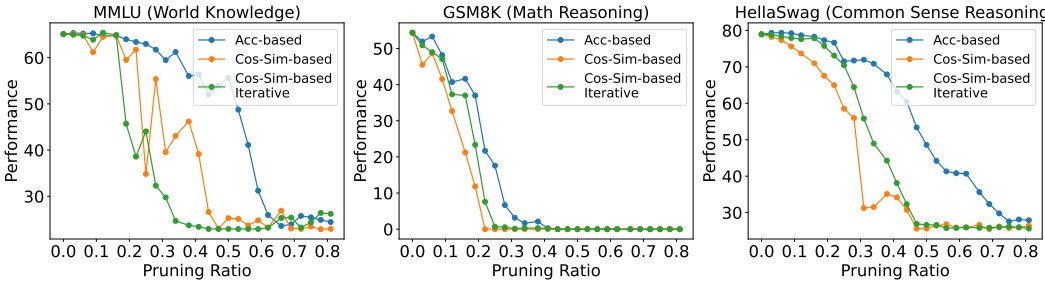

Figure 18: Evaluation of LLaMA-3-8B under the cosine-similarity pruning strategy of Gromov et al. (2025) compared with our proposed method and cosine-similarity score with an iterative pruning strategy.

## D  FURTHER ANALYSIS ABOUT DIFFERENCES BETWEEN TYPE OF TASKS

Figure 18 shows the same results as Figure 2, with the addition of results obtained using cosine similarity under an iterative pruning strategy. As discussed in the main paper, the original conclusions still hold—although the performance drop in HellaSwag is now less abrupt.

It's also worth noting that there are clear differences between the two ways of applying the cosine similarity score, which supports our argument that this metric should be used with caution when making assumptions about the internal mechanisms of Transformer models.

## E  STRUCTURED PRUNING

### E.1  IMPLEMENTATION DETAILS

Since all benchmarks used in our structured pruning experiments are multiple-choice tasks, we followed the same protocol and considerations as mentioned in Appendix C.1.

We evaluate models in a zero-shot setting on all tasks except for MMLU, where we use the five-shot format commonly adopted in prior work (Zhang et al., 2024b; He et al., 2024).

For Taylor relevance, we implement the element-wise importance formulation from Ma et al. (2023), using absolute weight–gradient products aggregated via sum—identified as the best-performing setup in their study. For Cosine Similarity, we follow the approach of He et al. (2024), concatenating multiple examples to form long input sequences that align with the model's context window. Explained in Appendix C.1.

All models are evaluated using the LM Evaluation Harness (Gao et al., 2024), ensuring consistency with prior structured pruning work. Experiments were run on NVIDIA RTX A6000 and RTX 4090 GPUs, using a batch size of 4.

### E.2  MISTRAL

To assess the generality of our approach across architectures, we replicate the structured pruning experiment described in the main paper (Section 6) using Mistral-7B. The pruning setup, datasets, evaluation method, and relevance proxies are identical to those used in the LLaMA3 experiment.

As shown in Table 4, our accuracy-based relevance method consistently outperforms all baselines across tasks, confirming its robustness beyond a single model family. However, unlike with LLaMA3, the task-specific pruned models do not surpass the performance of the unpruned model. This aligns with observations in prior work (He et al., 2024; Zhang et al., 2024b), which also report significant differences between models in the percentage of original performance retained after pruning.

Table 4: Accuracy of pruned Mistral-7B models on eight downstream tasks. All methods remove 25% of the layers using task-specific relevance estimates computed from each task's training set. Our accuracy-based approach consistently outperforms baselines. Best results per task are in bold. "Original" refers to the unpruned model.

| Methods | Arc-C | Arc-E | BoolQ | OBQA | HS | PIQA | WG | MMLU |
|---|---|---|---|---|---|---|---|---|
| Original | 53.67 | 79.55 | 83.73 | 44.4 | 81.03 | 82.26 | 74.27 | 62.48 |
| Taylor | 23.46 | 29.8 | 55.72 | 24.4 | 32.15 | 65.94 | 51.46 | 24.16 |
| Cosine Similarity | 42.41 | 60.23 | 66.73 | 37.4 | 70.43 | 73.72 | 70.48 | 42.29 |
| Out. Cosine-Sim | 38.41 | 68.39 | 64.62 | 37.8 | 70.82 | 78.56 | 61.8 | 26.69 |
| Out. Norm-Sim | 38.41 | 68.39 | 66.54 | 37.8 | 70.47 | 78.56 | 61.96 | 38.73 |
| Out. Divergence-Sim | 32.51 | 56.99 | 58.2 | 34.4 | 66.97 | 74.32 | 59.83 | 33.41 |
| Perplexity | 40.96 | 59.18 | 64.86 | 36.4 | 62.98 | 71.71 | 64.72 | 57.86 |
| Acc (Ours) | **46.42** | **74.83** | **82.29** | **42.4** | **75.77** | **80.52** | **72.46** | **61.18** |

Table 5: One-shot structured pruning results on LLaMA3-8B across eight downstream benchmarks. In this setting, relevance scores are computed once and used to prune 25% of layers in a single step. While our method occasionally underperforms others in this configuration, it remains highly competitive overall. Notably, the iterative version of our method consistently outperforms all one-shot baselines, highlighting the benefits of dynamic relevance estimation.

| Method | Arc-C | Arc-E | BoolQ | HS | OBQA | PIQA | WG | MMLU |
|---|---|---|---|---|---|---|---|---|
| Original | 53.16 | 81.02 | 82.02 | 78.94 | 44.8 | 81.28 | 73.56 | 65.11 |
| Taylor | 33.36 | 56.14 | 61.25 | 56.77 | 34.8 | 71.98 | 54.14 | 23.64 |
| Cosine Similarity | 47.61 | 68.86 | 70.4 | 71.09 | 39.4 | 76.39 | 70.39 | 35.12 |
| Out. Cosine-Sim | 44.8 | 68.01 | 56.33 | 51.19 | 38.2 | 73.29 | 59.19 | 23.72 |
| Out. Norm-Sim | 38.46 | 65.07 | 64.65 | 57.21 | 37.6 | 72.86 | 64.88 | 23.72 |
| Out. Divergence-Sim | 42.46 | 56.44 | 70.34 | 66.36 | 32.4 | 71.16 | 67.96 | 30.12 |
| Perplexity | 39.85 | 57.66 | 62.42 | 55.05 | 37.2 | 66.81 | 65.59 | 59.63 |
| Acc 1-Shot (Ours) | 42.24 | 72.09 | 52.2 | **74.49** | **44.4** | **79.54** | 66.93 | 53.21 |
| Acc Iterative (Ours) | **49.57** | **74.96** | **84.04** | 71.53 | 44 | 79.06 | **73.8** | **62.97** |

### E.3 ONE-SHOT

To assess how our method performs in a simpler pruning setup, we replicate the main structured pruning experiment using a one-shot approach. Instead of iteratively updating relevance scores during pruning, we compute each method's scores only once, rank the layers accordingly, and prune the bottom 25% in a single step.

Results are shown in Table 5. While our method occasionally underperforms others in the one-shot setting (e.g., on BoolQ), the iterative version of our method still outperforms all base-lines—including one-shot variants—highlighting the benefits of reevaluating relevance dynamically. This is consistent with our earlier findings in Section C.5, where we showed that block relevance evolves during pruning.

Interestingly, for a few datasets (e.g., HellaSwag and OpenBookQA), our one-shot variant marginally outperforms its iterative counterpart. We hypothesize that this may result from domain shifts between the training and test splits, which can affect our accuracy-based signal. Additionally, selecting the optimal pruning set is ultimately a challenging search problem—one that has been tackled explicitly in recent works (Sieberling et al., 2024).

### E.4 TASK-INDEPENDENT PRUNING

The task-independent structured pruning setup consists of using a single dataset—commonly referred to as a calibration dataset—to compute relevance scores and prune the model accordingly. This results in one pruned model per pruning method, which is then evaluated across multiple down-stream tasks. The tasks used for evaluation typically mirror those presented in the main paper.

It is worth noting that there is no standardized protocol regarding which dataset to use as calibration data or how many samples to include. For example, Zhang et al. (2024b) uses WikiText-2 with 10 randomly selected instances, while He et al. (2024) uses C4, selecting 256 samples where each sample may span multiple instances (due to concatenation to match the model's input length; see Appendix C.1).

In our experiments, we adopt the setup of He et al. (2024) for consistency. However, to ensure a fair comparison—especially against pruning methods like cosine similarity that operate on instance-level granularity—we avoid concatenation and instead use the same 1,500 instances employed in the cosine similarity baseline. These instances were selected to construct 256 full-context-length samples in a consistent and comparable manner.

Table 6 presents results for the LLaMA3-8B model under this classic pruning setup. As shown, the cosine similarity method outperforms all others, including our accuracy-based metric. This outcome contrasts with the results reported by Zhang et al. (2024b), likely due to differences in calibration dataset choice and sample size.

These results are consistent with our expectations. Our method is tightly coupled to the calibration dataset, and—as demonstrated throughout this paper—relevance is highly task- and data-dependent. Therefore, when the calibration dataset is misaligned with the evaluation tasks, performance is likely to degrade.

Then, we evaluate our method using an alternative calibration dataset that combines training data from the target evaluation tasks. Table 2 reports results for LLaMA3-8B using a calibration set composed of 10% of the training data from each of the eight benchmarks. Under this configuration, our method outperforms all baselines, yielding a pruned model that achieves the highest average performance across tasks.

### E.5 TASK RELATIONS

Given the task-independent results presented in Table 2, a natural question arises: can the training set of one task serve as a suitable calibration set for pruning models used in other tasks? Table 7 explores this by showing the performance of different training sets used as calibration data. We observe that most tasks achieve good average performance; notably, some tasks serve as particularly effective calibration sets for others.

Table 6: Task-independent structured pruning results for LLaMA3-8B across eight downstream benchmarks. Each pruning method uses the same 1,500-instance calibration dataset to prune the model once, which is then evaluated on all tasks. Cosine similarity performs best in this setup, while our accuracy-based method underperforms, likely due to its strong dependency on the calibration dataset.

|  | Arc-C | Arc-E | BoolQ | HS | OBQA | PIQA | WG | MMLU | Mean |
|---|---|---|---|---|---|---|---|---|---|
| Original | 53.15 | 81.02 | 82.02 | 78.94 | 44.8 | 81.28 | 73.56 | 65.11 | 69.99 |
| Taylor | **45.39** | 67.97 | 61.31 | 63.73 | **41.4** | 76.55 | 68.11 | 25.03 | 56.19 |
| Cosine Similarity | 43.34 | 65.32 | **76.7** | **70.24** | 36.8 | 73.39 | **70.96** | **40.78** | **59.69** |
| Out. Cosine-Sim | 44.2 | **72.05** | 71.99 | 66.57 | 40.2 | 77.37 | 66.93 | 34.56 | 59.23 |
| Out. Norm-Sim | 42.66 | 70.12 | 66.94 | 67 | **41.4** | **78.73** | 67.72 | 34.1 | 58.58 |
| Out. Divergence-Sim | 43.54 | 71.25 | 68.62 | 65.09 | 39.6 | 76.01 | 65.94 | 30.57 | 57.58 |
| Perplexity | 40.19 | 63.47 | 44.43 | 65.96 | 39.2 | 74.48 | 64.4 | 29.4 | 52.69 |
| Acc (Ours) | 36.69 | 56.52 | 53.36 | 60.16 | 33.8 | 72.63 | 60.14 | 28.5 | 50.23 |

Table 7: Calibration dataset analysis. Each row shows the performance of a LLaMA3-8B model pruned at 25% with our method using a different train set as a calibration dataset.

|  | Arc-C | Arc-E | BoolQ | HS | OBQA | PIQA | WG | MMLU | Mean |
|---|---|---|---|---|---|---|---|---|---|
| Arc-C | 49.57 | 74.45 | 69.08 | 72.91 | 42.2 | 77.8 | 67.56 | 40.2 | 61.72 |
| Arc-E | 51.37 | 74.96 | 66.94 | 73.62 | 43.6 | 78.51 | 71.59 | 44.82 | 63.18 |
| BoolQ | 40.7 | 66.96 | 84.1 | 67.97 | 38.6 | 73.23 | 71.82 | 35.09 | 59.81 |
| HS | 44.45 | 62.5 | 65.84 | 71.53 | 44.2 | 73.5 | 64.72 | 42.83 | 58.7 |
| OBQA | 45.82 | 66.5 | 75.38 | 66.88 | 44 | 75.24 | 65.9 | 50.45 | 61.27 |
| PIQA | 44.62 | 70.2 | 48.62 | 68.45 | 44.2 | 79.05 | 67.8 | 27.8 | 56.34 |
| WG | 44.71 | 68.73 | 78.13 | 69.85 | 39.2 | 75.03 | 73.8 | 42.4 | 61.48 |
| MMLU | 40.61 | 62.46 | 75.32 | 63.11 | 35 | 71.49 | 69.3 | 62.97 | 60.03 |

Building on Table 7, Figure 19 presents a graph illustrating the relationships between tasks. Each node corresponds to a task, and a directed edge from task 1 to task 2 indicates that the training set of task 1 serves as either a good (blue) or poor (red) calibration set for task 2. We define "good" as achieving at least 90% of the performance obtained when pruning with task 2's own training set (see Table 1), and "poor" as 10% or lower. To further indicate the strength of the effect, edge transparency varies: lighter blue denotes values closer to the 90% threshold, while lighter red denotes values closer to 10%.

Several observations emerge from this analysis:

- ARC-E and ARC-C serve as good proxies for almost all tasks, with the exception of BoolQ and MMLU. Interestingly, ARC-E is a better proxy than ARC-C, despite being an easier version of the same benchmark.

- Nearly all tasks act as good proxies for HellaSwag, except for MMLU. This finding is noteworthy because HellaSwag is generally considered a commonsense reasoning task, whereas MMLU requires broader world knowledge.

- No task provides a good proxy for MMLU or BoolQ. For MMLU, this is expected: as a world-knowledge benchmark, it likely requires calibration sets with overlapping domain coverage, which the other tasks lack. For BoolQ, however, the absence of good proxies is less straightforward. One possible explanation is that its yes/no format introduces unique structural properties that are particularly sensitive to pruning.

E.6 COMPUTATIONAL COST

To assess how long our method would take to achieve higher pruning ratios, quantify the benefits of using more compute, and evaluate larger models, we estimate the time required to prune 50% of LLaMA-3-8B using an NVIDIA L40S and a pair of NVIDIA H100s, as well as the time required

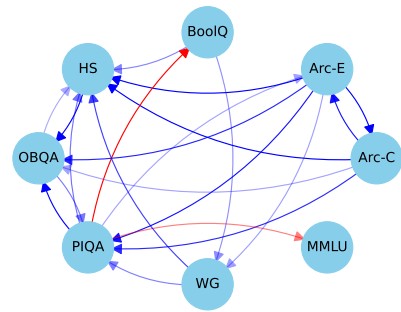

Figure 19: Relation between tasks. Computed from data in Table 7

Table 8: Estimated time (hours or days) required for pruning 50% of the model using our method using 1,000 instances per dataset.

| Model | Batch Size | Hardware | C4 | LIMA | MathInstruct | CodeAlpaca |
|---|---|---|---|---|---|---|
| Llama 3-8B | 8 | 1 × L40s | 11.56 hrs | 10.02 hrs | 4.72 hrs | 1.60 hrs |
| Llama 3-8B | 64 | 2 × H100 | 7.46 hrs | 4.68 hrs | 3.23 hrs | 0.83 hrs |
| Llama 3-70B | 8 | 2 × H100 | 8.49 days | 6.85 days | 3.39 days | 1.12 days |

to prune 50% of LLaMA-3-70B using two NVIDIA H100s. These estimates are derived from the timings reported in Table 3, along with additional inference runs performed on both models using the dual-H100 setup. For each dataset, we measured the runtime using the maximum feasible batch size and recorded the reduction in runtime obtained after removing a block. The resulting estimates, summarized in Table 8, show that additional compute substantially benefits our method and that pruning to higher ratios—even for larger models—is feasible.

Additionally, we analyze how runtime scales with the size of the calibration dataset. Figure 20 shows the time per sample for different calibration-set sizes when pruning 25% of LLaMA-3-8B on an NVIDIA L40S. The results indicate that, beyond approximately 250 samples, the time per sample becomes stable and even slightly decreases across all datasets. This suggests that, in the worst case, our method scales linearly with the number of calibration samples.

Reducing the computational cost of our approach remains an important direction for future work. Parallelizing relevance computation across multiple GPUs—for example, assigning different subsets of layers to each device—could substantially reduce runtime. Additional gains may come from inference-optimized frameworks or quantization, though the latter may affect pruning behavior and requires further study. Moreover, many optimized frameworks do not yet support model modification, limiting their applicability to our method.

### E.7 HEALING

Given that our results so far have not used healing as a post-processing step, a natural question arises: Could healing allow less computationally expensive methods, such as Cosine Similarity, to achieve comparable performance? Moreover, are the blocks selected by our method truly less important for the task than those selected by other methods?

To address these questions, we applied a healing process to our task-dependent setup using the training split of each benchmark (the same data used for calibration during pruning). Tables 9, 10, and 11 show a comparison of our method, Cosine Similarity, and the selection of random blocks to prune the same 8 benchmarks used in previous sections. After pruning at a 25% ratio and then healing (Table 9), our method performs similarly to Cosine Similarity. However, we do not interpret this as evidence that our method is selecting worse—or even equally important—blocks compared to Cosine Similarity. Instead, we believe this behavior reflects the fact that a LLaMA-3-8B pruned by

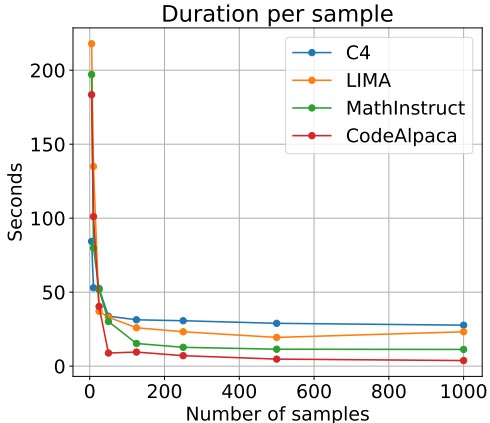

Figure 20: Time per sample versus the number of calibration samples for our method. Results are shown across multiple datasets using an L40S GPU while pruning 25% of LLaMA-3-8B.

25% remains expressive enough to perform well on these tasks. Supporting this interpretation, we observe that even randomly selecting blocks to prune, followed by healing, can sometimes achieve competitive results.

We repeated the same experiment at pruning ratios of 50% (Table 10) and 75% (Table 11). At 50% pruning, our method consistently outperforms Cosine Similarity even after healing. At 75%, it still outperforms the baselines in several cases. The cases where our method no longer leads, however, coincide with performance levels close to random selection, suggesting that the model simply lacks sufficient parameters to solve the task at such extreme sparsity.

A similar trend—where healing provides clear benefits over other pruning methods only around the 50% pruning regime—was also reported by (Gromov et al., 2025), who compared Cosine Similarity with an even simpler pruning method across multiple pruning ratios and tasks.

To implement the healing stage, we fine-tuned each pruned model for 10 epochs using the corresponding training set for each task (see Figure 21 for per-epoch performance). For each method–task pair, the tables report the best-performing epoch within this window. Across all experiments, model performance consistently peaked during the 10-epoch schedule and then began to decline, indicating the onset of overfitting to the training set.

Following prior work (Gromov et al., 2025), we employed the Hugging Face `Trainer API` (Wolf et al., 2020), QLoRA quantization using the `bitsandbytes` library (Dettmers et al., 2023), and LoRA adapters (Hu et al., 2022) implemented with the `peft` library (Mangrulkar et al., 2022). The fine-tuning configuration was as follows:

- Applied modules: `[gate_proj, down_proj, up_proj]`
- Batch size: 16
- LoRA $\alpha$: 2
- LoRA rank: 2
- Peak learning rate: 3e-4
- LoRA dropout: 0.05
- LR scheduler: cosine
- Warmup steps: 100

Table 9: Results on the 8 tasks after pruning **25%** of the model LlaMa-3-8B using our method, cosine similarity, or random block pruning, we show the results with and without healing stage.

| Method | Healing | Arc-C | Arc-E | BoolQ | HS | OBQA | PIQA | WG | MMLU |
|---|---|---|---|---|---|---|---|---|---|
| Accuracy (Ours) | No | 49.57 | 74.96 | 84.04 | 71.53 | 44.00 | 79.06 | 73.80 | **62.97** |
| | Yes | **57.42** | **83.84** | **89.30** | 75.71 | 51.20 | **84.22** | 82.08 | 61.64 |
| Cosine Similarity | No | 45.73 | 67.80 | 66.33 | 69.52 | 38.60 | 72.91 | 71.35 | 44.05 |
| | Yes | 56.48 | 82.07 | 88.99 | **76.08** | **53.80** | 81.66 | **83.03** | 58.76 |
| Random | No | 24.08 | 36.80 | 46.03 | 45.74 | 25.48 | 51.73 | 44.73 | 25.47 |
| | Yes | 43.86 | 73.68 | 82.16 | 67.06 | 46.84 | 78.68 | 74.65 | 34.63 |

Table 10: Results on the 8 tasks after pruning **50%** of the model LlaMa-3-8B using our method, cosine similarity, or random block pruning, we show the results with and without healing stage.

| Method | Healing | Arc-C | Arc-E | BoolQ | HS | OBQA | PIQA | WG | MMLU |
|---|---|---|---|---|---|---|---|---|---|
| Accuracy (Ours) | No | 28.67 | 40.95 | 79.57 | 48.69 | 32.40 | 63.28 | 54.46 | **55.64** |
| | Yes | **38.65** | **62.96** | **86.33** | **57.39** | **40.60** | **75.24** | **70.48** | 55.53 |
| Cosine Similarity | No | 25.34 | 27.78 | 42.72 | 26.64 | 29.40 | 52.77 | 50.51 | 22.95 |
| | Yes | 33.96 | 59.68 | 78.38 | 51.31 | 40.40 | 73.01 | 66.14 | 26.11 |
| Random | No | 20.87 | 19.72 | 42.24 | 26.52 | 22.36 | 41.14 | 39.43 | 24.85 |
| | Yes | 26.96 | 52.00 | 67.06 | 41.55 | 34.12 | 68.78 | 59.16 | 25.67 |

Table 11: Results on the 8 tasks after pruning **75%** of the model LlaMa-3-8B using our method, cosine similarity, or random block pruning, we show the results with and without healing stage.

| Method | Healing | Arc-C | Arc-E | BoolQ | HS | OBQA | PIQA | WG | MMLU |
|---|---|---|---|---|---|---|---|---|---|
| Accuracy (Ours) | No | 26.11 | 28.66 | 62.17 | 27.56 | 26.80 | 55.66 | 50.51 | 25.46 |
| | Yes | 25.09 | **40.91** | **63.06** | **30.43** | **31.00** | **65.02** | **51.70** | **25.83** |
| Cosine Similarity | No | **27.30** | 26.56 | 57.13 | 26.05 | 28.00 | 51.80 | 50.12 | 24.32 |
| | Yes | 26.19 | 34.60 | 63.00 | 27.48 | 29.40 | 61.59 | 50.83 | 24.28 |
| Random | No | 20.90 | 20.39 | 37.99 | 26.50 | 22.80 | 40.97 | 40.68 | 24.32 |
| | Yes | 24.78 | 38.57 | 61.11 | 28.50 | 28.00 | 61.10 | 51.74 | 25.22 |

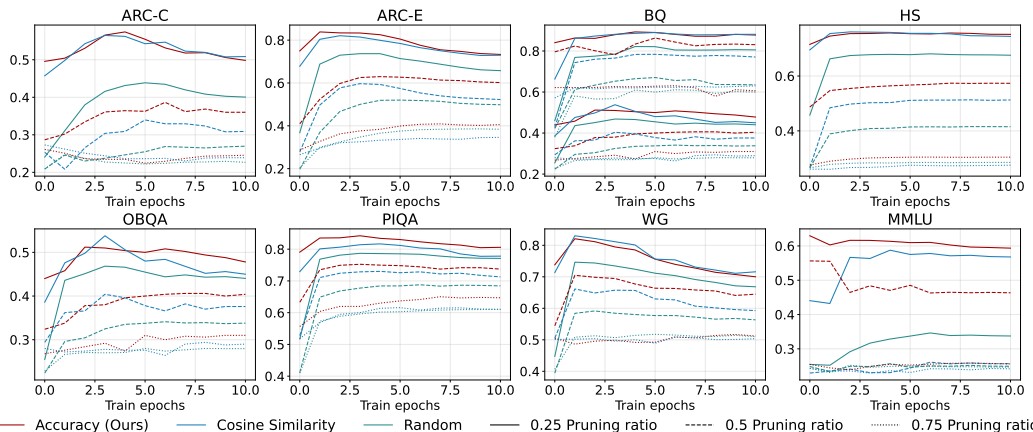

Figure 21: Impact of healing after pruning across varying pruning ratios and train epochs.

