# OpenReview forum: "Rethinking Layer Relevance in Large Language Models Beyond Cosine Similarity"
_ICLR.cc/2026/Conference — ICLR 2026 Poster_

### Official Review · Reviewer_5sPj · 2025-10-15

**Soundness:** 2
**Presentation:** 2
**Contribution:** 3
**Rating:** 4
**Confidence:** 3

**Summary:**

This paper critiques cosine similarity as a metric for layer relevance in transformers and proposes an accuracy-based alternative that directly measures performance degradation from layer removal. The authors provide theoretical proof that cosine similarity can arbitrarily misestimate layer importance and demonstrate their accuracy-based metric achieves better results in structured pruning tasks.

**Strengths:**

- The theoretical contribution (Theorem 1) formally proves that layers with arbitrarily low cosine similarity can still be critical for performance, providing rigorous justification for questioning this widely-used metric. The proof construction cleverly exploits the "snowball effect" where subtle layer modifications are amplified by downstream layers, offering valuable theoretical insight into transformer dynamics.
  - The empirical evaluation is thorough, covering multiple models (LLaMA3-8B, Mistral-7B, OLMo) and eight diverse benchmarks, with careful ablations comparing iterative versus one-shot pruning and task-dependent versus task-independent settings. The visualizations effectively illustrate the discrepancies between cosine similarity and actual relevance, particularly the striking example of OLMo's layer 16.

**Weaknesses:**

- The computational cost of the proposed accuracy-based metric is a major practical limitation that receives insufficient treatment. While Section 6.1 mentions the method requires N × T forward passes compared to T forward passes for cosine similarity (where N is the number of layers), the paper lacks concrete wall-clock time comparisons, memory requirements, or strategies to make the approach tractable for large-scale models. For a 70B parameter model with 80 layers, computing layer-by-layer relevance becomes prohibitively expensive. The brief mention of "parallel computation" and future work on cost reduction does not adequately address whether this metric can realistically replace cosine similarity in practice.
  - The assumption that accuracy on a calibration dataset is the "ground truth" for layer relevance may itself be problematic in ways not fully explored. The paper acknowledges strong sensitivity to calibration data choice (Table 6 shows performance varying from 56.34% to 63.18% depending on which task's training set is used), yet does not investigate: (1) whether small calibration sets produce stable relevance estimates, (2) how relevance rankings change with different calibration sizes, (3) whether the metric captures generalizable layer properties or just task-specific overfitting patterns. The task-independent results (Table 5) where cosine similarity outperforms the accuracy metric (59.69% vs 50.23%) raise questions about which metric truly captures layer importance.
  - The connection between the theoretical worst-case construction (Theorem 1) and practical transformer behavior remains unclear. The proof constructs a contrived 3-layer model that perfectly overfits a dataset, but the paper does not demonstrate that similar pathological patterns actually occur in real trained transformers. Missing are analyses showing: (1) whether real transformer layers exhibit the "snowball effect" mechanism from the proof, (2) what proportion of layers in practice have low cosine similarity but high accuracy-based relevance, or (3) quantitative measures of the correlation between the two metrics across different model families and training stages. The empirical sections jump from the theorem to pruning experiments without bridging this gap.
  - The evaluation methodology has potential confounds that could inflate the apparent superiority of the accuracy-based metric. In the task-dependent setting (Table 1), the calibration data is the training set of the target task, which may give the accuracy metric an unfair advantage since it directly optimizes for performance on data from the same distribution. A more rigorous evaluation would use separate calibration and evaluation splits from each task. Additionally, the paper does not control for the possibility that the accuracy metric simply overfits to the calibration data distribution. The finding that some layers have negative relevance (green in Figure 1A) — meaning accuracy improves upon their removal — suggests the metric may be capturing dataset-specific artifacts rather than fundamental layer importance. I will reconsider my score in the rebuttal.

**Questions:**

see weaknesses

---

> ### Author Response · Authors · 2025-12-03
>
> We thank the reviewer for the constructive feedback and for explicitly noting their willingness to reconsider the evaluation.
>
> The strengths highlighted in the review align closely with our core contributions. Theorem 1 provides a formal and constructive proof that layers with arbitrarily low cosine similarity can still be critical for model performance. This result offers a rigorous foundation for questioning the widespread reliance on cosine similarity as a proxy for layer relevance and illustrates how small perturbations can accumulate and propagate through downstream layers.
>
> We also appreciate the reviewer’s recognition of the breadth and rigor of our empirical evaluation, as well as the positive comments on our visualizations. These clearly demonstrate the divergence between cosine similarity and true relevance—most notably the striking mismatch observed in OLMo’s layer 16—underscoring the practical significance of our findings.
>
> The review raises four concerns: (1) Computational cost, (2) Role of calibration dataset, (3) Connection between theory and practice, and (4) Potential confounds in evaluation. We address each below.
>
> ---
> **1. Computational Cost**
>
> We now include wall-clock measurements for pruning 25% of LLaMA3-8B across four datasets on identical hardware (Table 3). While our method is slower than cosine similarity (4.6 hours vs. 8.64 minutes), it is comparable to other baselines and far from the slowest (Taylor: 10.37 hours).
>
> To address scalability, we provide worst-case estimates for pruning 50% of LLaMA3-70B (Table 8):
> - C4: ~8.5 days
> - CodeAlpaca: ~1.1 days
>
> Section 6.1 and Appendix E.6 now include an expanded discussion of these results and outline practical avenues for reducing cost, including quantization, optimized inference runtimes, and full parallelization of layer evaluations.

---

> ### Author Response · Authors · 2025-12-03
>
> **2. Role of Calibration Dataset**
>
> The reviewer raised three questions:
> - (1) Stability with small calibration sets,
> - (2) Sensitivity to calibration size, and
> - (3) Whether the metric captures generalizable properties or task-specific patterns.
>
> For (1) and (2), we added new experiments on calibration-set size (Appendix C.6). These show that relevance scores stabilize at ~500 samples, indicating that relatively small calibration sets can yield reliable estimates.
>
> For (3), our original submission included cross-task calibration experiments: pruning with one task and evaluating across all others. These results (Section 6, Table 2; Appendix E.5) show that our metric does generalize—pruning with one dataset can still yield strong performance on others.
>
> However, we emphasize that layer relevance is inherently task-dependent, which aligns with our interpretability perspective. For example, achieving strong performance on MMLU requires a calibration set with world-knowledge coverage; datasets lacking this information may prune away critical blocks. As shown in Appendix E.5, none of the other task training sets reliably calibrates MMLU, underscoring this dependence.
>
> ---
> **3. Connection Between Theory and Practice**
>
> We thank the reviewer for the insightful feedback on linking the theoretical construction to real transformer behavior. The reviewer raised three points:
> - (1) whether real layers exhibit the “snowball effect” mechanism from the proof,
> - (2) what proportion of layers have low cosine similarity but high accuracy-based relevance, and
> - (3) quantitative measures of correlation between the two metrics across models and training stages.
>
> Points (2) and (3): These are addressed by analyses already in the paper.
> - **Figures 1 and 4**: Heatmaps showing cosine similarity vs. accuracy-based relevance across two model families and multiple tasks, illustrating clear divergence.
> - **Figure 3A**: Correlation analysis between the metrics.
> - **Figure 3B**: Mismatch ranking, highlighting layers with low cosine similarity but high relevance (and vice versa).
>
> Together, these provide quantitative evidence of the proportion of mismatched layers and the overall correlation structure, directly addressing the reviewer’s concerns.
>
> Point (1): We agree that identifying the “snowball effect” in large trained transformers is challenging. However, the theorem’s purpose is conceptual: it establishes that cosine similarity can fail under specific conditions, providing theoretical motivation for our empirical investigation. This connection strengthens the case for moving beyond heuristic metrics toward accuracy-grounded relevance measures.
>
> ---
> **4. Potential Confounds in Evaluation**
>
> The reviewer raised two concerns:
> - (1) whether using the target task’s training set as calibration data gives our metric an unfair advantage, and
> - (2) whether negative relevance values indicate artifacts rather than meaningful behavior.
>
> **Calibration fairness:**
>
> All methods in each table are evaluated under identical calibration conditions. In the task-dependent setting (Table 1), every pruning method—including cosine similarity—uses the same calibration dataset. This ensures a fair comparison that isolates the effect of the metric itself, not the data it receives. Importantly, our evaluation explicitly tests generalization: calibration and test sets are always disjoint, so relevance estimates are computed on one dataset and evaluated on separate test data, preventing any overlap.
>
> To further address this concern, we include cross-task experiments (Appendix E.5, summarized in Section 6), where pruning is performed using one task’s training set and evaluated across other tasks. These results show that our accuracy-based metric generalizes well, often achieving strong performance even when calibration and evaluation distributions differ. This directly counters the concern that the metric simply overfits to the calibration dataset.
>
> **Negative relevance values:**
>
> The reviewer noted green cells in Figure 1A. These are not artifacts but reflect a real property of the model: layer relevance is task-dependent. Some layers encode features that help certain distributions but harm others. Our metric exposes these cases, revealing when layers introduce biases that reduce accuracy on specific tasks. This insight connects directly to our mechanistic interpretability contribution—accuracy-based relevance uncovers behaviors that heuristic metrics like cosine similarity cannot.

---

### Official Review · Reviewer_a5Py · 2025-10-28

**Soundness:** 2
**Presentation:** 3
**Contribution:** 2
**Rating:** 4
**Confidence:** 3

**Summary:**

The paper addresses how to assess layer relevance in large language models (LLMs) and challenges the widespread use of cosine similarity between layer inputs and outputs as a proxy for layer importance. Through a theoretical worst-case analysis and broad empirical evidence, the authors show that cosine similarity can fail to capture true layer relevance. They propose an alternative metric based on the actual drop in model accuracy after layer removal and demonstrate its advantages for structured pruning.
However, the practical feasibility of the proposed approach and certain implementation details of the numerical comparisons remain unclear. If these concerns were addressed, the paper would make a meaningful contribution to understanding layer relevance in large models and is likely to stimulate further work in this area.

**Strengths:**

The paper studies a widely used heuristic (cosine similarity) as a proxy for layer relevance, and demonstrates its unreliability. This would be an important contribution given the prevalence of cosine-based analysis in interpretability work. The paper is clearly written and well organized, making the arguments easy to follow.

**Weaknesses:**

W1. **Computational practicality**

The proposed accuracy-drop metric, while conceptually sound, is computationally intensive, as it requires re-evaluating model performance after removing each layer. However, the paper provides no quantitative assessment of this cost relative to cosine similarity or other baselines. Without such analysis, readers are left uncertain whether the approach is feasible for large-scale pruning or limited to research-scale evaluation.

W2. **Effect of layer removal**

It is unclear whether the observed accuracy drops truly reflect layer relevance or artifacts of the chosen pruning method. The authors note that "no healing or post-processing was applied." This raises the possibility that the measured performance degradation arises partly from architectural disruption rather than intrinsic irrelevance. Since cosine similarity ignores activation magnitude, a pruning operator without residual rescaling can make cosine-guided deletions appear disproportionately harmful. Therefore, the claim that "cosine-based pruning drops performance more" may hold only for some pruning methods and might not generalize to pruning frameworks that include rescaling or post-hoc recovery steps.

If the authors can address the two main concerns, I would be happy to raise my overall score.

**Questions:**

Q1. Could you quantify the computational cost of the proposed accuracy-drop metric relative to cosine similarity and other baselines?

Q2. Have you tested alternative ablation mechanisms? If not, could differences in ablation strategy explain part of the observed performance drops?

Q3. Relatedly, could the apparent weakness of cosine-guided pruning partly arise because cosine ignores magnitude, and your ablation protocol does not re-normalize downstream layers? In that case, the observed degradation might reflect an uncompensated magnitude mismatch rather than a true failure of cosine similarity.

---

> ### Author Response · Authors · 2025-12-03
>
> We thank the reviewer for the thoughtful and constructive summary of our work and for explicitly noting their willingness to increase their evaluation if we addressed the two concerns—which we did.
>
> We appreciate the clear recognition of our main contribution: rigorously evaluating cosine similarity, a widely used but rarely questioned proxy for layer relevance, and demonstrating its unreliability through both theory and extensive empirical evidence. Given how pervasive cosine-based analyses are in mechanistic interpretability, we are encouraged that the reviewer sees the importance and potential impact of this direction. We also value the positive assessment of the paper’s clarity and organization—presenting the arguments in an accessible and coherent way was a central goal.
>
> The review raises two concerns: (1) Computational cost and (2) Effect of layer removal. We address each below.
>
> ---
> **1. Computational Cost**
> In the revised manuscript, we include extensive runtime analysis for all methods and examine scalability to larger models and varying calibration set sizes (Section 6.1, Appendix E.6).
> As noted in the original submission, our method is faster than prior pruning approaches (e.g., perplexity and output-based methods) but slower than cosine similarity, which is extremely fast but—as we show—is unreliable for measuring layer relevance in LLMs.
>
> Mean wall-clock times for pruning 25% of LLaMA3-8B across four datasets (Table 3):
> - Accuracy (ours): 4.6 hours
> - Cosine Similarity: 8.6 minutes
> - Perplexity: 4.6 hours
> - Output Cos-Sim / Norm: 4.8 hours
> - Output Divergence: 4.9 hours
> - Taylor: 10.4 hours
>
> For LLaMA3-70B at 50% pruning, runtime ranges from ~8.5 days (C4) to ~1.1 days (CodeAlpaca) (Section 6.1; Appendix E.6). We also show linear scaling with calibration set size (Figure 20).
>
> These runtimes should be interpreted as upper bounds. Our relevance computation is inherently parallelizable, and future work can further reduce cost via quantization and optimized inference runtimes.
>
> ---
> **2. Effect of Layer Removal**
>
> The reviewer raised an interesting hypothesis:
>
> > “_It is unclear whether the observed accuracy drops truly reflect layer relevance or artifacts of the chosen pruning method… post-hoc recovery steps might change the outcome._”
>
> To address this, we ran new experiments using healing (a common post-processing technique). Results show that the reviewer’s hypothesis does not hold:
> - Our accuracy-based metric still outperforms cosine similarity even when healing is applied.
> - These results are included in Section 6.2 and detailed in Appendix E.7.
>
> **Key findings**:
> - At low pruning ratios (25%), the performance gap narrows after healing. However, this does not indicate that cosine similarity selects better layers. Instead, it reflects that a model with only 25% of layers removed remains expressive enough to recover performance. Supporting evidence:
>     - Even random pruning achieves competitive post-healing performance at 25%.
>     - Prior work reports similar behavior for simple baselines.
> - At higher pruning ratios (50–75%), our method significantly outperforms cosine similarity, even with healing applied.
>
> ---
> **Specific Questions**
>
> **Computational cost relative to baselines?**
> Included in Table 3. Summary: pruning 25% of LLaMA3-8B takes 4.6 hours for our method vs. 8.6 minutes for cosine similarity. Our method is the second fastest overall; Taylor is slowest at 10.37 hours.
>
> **Alternative ablation mechanisms?**
> Yes. We explored one-shot vs. iterative pruning, task-dependent vs. task-independent setups, healing vs. no healing, and multiple calibration datasets. Our method outperforms all baselines in nearly all cases, except certain task-independent configurations where calibration data lacks required knowledge.
>
> **Could cosine’s weakness arise from ignoring magnitude or lack of normalization?**
> No. Our original experiments normalized inputs to each layer, and the new healing experiments confirm that our method still outperforms cosine similarity. These results eliminate the possibility that degradation is due to missing normalization or post-hoc adjustment.

---

### Official Review · Reviewer_BfoK · 2025-10-30

**Soundness:** 3
**Presentation:** 3
**Contribution:** 3
**Rating:** 4
**Confidence:** 4

**Summary:**

This work discusses the issues of layer removal based on cosine similarity and proposes an alternative method which directly measures the impact of layer removal on the downstream performance.

**Strengths:**

- Good motivation. Directly measuring the impact of removing layers on accuracy is ideal.
- Outperforms prior methods.
- Relatively simple.

**Weaknesses:**

- The main issue I have with the method is that it only measures the impact of removing a single layer. In the scenario where multiple layers are removed, the metric does not identify which layers are dependent to one another. The authors state: "our method prunes blocks iteratively, re-evaluating the model after each step". This can result in the removal of a layer which on its own does not result in any drastic drop in performance, but the joint removal of this layer and a subsequent cause drastic drop in performance. In contrast, there could be another layer which if removed first causes a greater drop in performance, but other layers not being as dependent on it, further removals might result in a lesser total performance drop. This is not something that is addressed in the paper, even just in discussion.
- As the authors mentioned themselves "when the calibration set is restricted to a single benchmark, performance varies significantly". Hence this method might not be applicable in scenarios where the downstream application is unknown or very general.
- The computational cost of different methods are reported as equations, but actual numbers in the tables would help.

**Questions:**

- line 15: "On this work," -> "In this work,"
- Line 294 "a random baseline in the dataset" is not the clearest way to express that the model is a random predictor. I assume that this is what the authors meant.

---

> ### Author Response · Authors · 2025-12-03
>
> We thank the reviewer for the thoughtful feedback and for recognizing key strengths of our work: the motivation for directly measuring the accuracy impact of layer removal, the empirical improvements over prior relevance metrics, and the simplicity of the proposed approach. Our goal was precisely to develop a method that is conceptually straightforward yet faithful to what layer relevance truly reflects—its contribution to downstream task performance—and we are glad to see this acknowledged.
>
> Beyond pruning performance, we would like to emphasize the method’s value for mechanistic interpretability. A central challenge in this area is determining which components of a transformer genuinely matter for model behavior, and there is currently no consensus on how to measure layer relevance. Prior work has often relied on cosine similarity or representation-based heuristics, but our findings show that these proxies can lead to questionable conclusions—including the claim that block relevance is task-independent. By grounding relevance in actual performance impact, our metric offers a more reliable diagnostic tool that can help the ML community better understand model structure, analyze task-dependent behavior, and even identify layers whose removal unexpectedly improves accuracy.
>
> The reviewer raises four concerns: (1) Search-based pruning, (2) Generalization to unknown tasks, (3) Computational cost, and (4) Minor wording questions. We address each below.
>
> ---
>
> **Search-Based Pruning**
>
> The reviewer suggests that pruning could be improved by finding the optimal combination of n layers to remove. While this is true in principle, the computational complexity is prohibitive: for example, pruning 3 layers from a 32-layer model requires evaluating 32×31×30 = 29,760 combinations, compared to only 93 evaluations with our greedy approach—over 300× slower. This limitation is common across pruning methods, which universally rely on iterative, greedy removal steps for tractability.
>
> To explore this, we conducted a small brute-force experiment on LLaMA-3-8B using CodeAlpaca, exhaustively evaluating all pairs of layers. Interestingly, the optimal pair matched the layers selected by our iterative procedure. While promising, this result is limited to one dataset and two-layer subsets, so we did not include it in the paper. In the revised manuscript (Section 6.1), we added a discussion of this point and highlighted it as a direction for future work, where pruning the search space will be key to making optimal selection feasible.
>
> ---
>
> **Generalization to Unknown Tasks**
>
> The reviewer notes that our method might not apply when the downstream application is unknown or very general. Our paper shows that task-independent performance varies with the calibration set, but we also include cross-task calibration experiments where pruning is done using one task and evaluated across all others. These results (Section 6, Table 2; Appendix E.5) demonstrate that our metric does generalize across tasks, and thus can be applicable in such scenarios.
>
> As future work, we plan to further study the relationship between calibration-set composition and task-independent performance. So far, the best results occur when the calibration set includes examples from diverse tasks, though we also observed cases where a single-task calibration set produced strong generalization.
>
> ---
>
> **Computational Cost**
>
> We appreciate the suggestion to report computational times. The revised version now includes wall-clock times for pruning 25% of LLaMA-3-8B across multiple datasets using identical hardware for all methods, as well as worst-case estimates for pruning 50% of LLaMA-3-70B. These results are presented in Section 6.1 and Table 3, and detailed further in Appendix E.6 and Table 8.
>
> ---
> **Minor Wording Questions**
>
> Thank you for pointing these out. We have corrected the phrasing in the corresponding lines.

---

### Official Review · Reviewer_J9R4 · 2025-11-02

**Soundness:** 3
**Presentation:** 3
**Contribution:** 3
**Rating:** 6
**Confidence:** 4

**Summary:**

This paper investigates whether cosine similarity between a layer's input and output is a reliable proxy for that layer's relevance in large language models. The authors prove a theoretical worst case showing a layer can have arbitrarily low cosine-similarity yet be crucial to performance because small changes it introduces are amplified downstream. They then empirically compare cosine-similarity rankings to a ground-truth relevance obtained by measuring the actual drop in task accuracy after ablating layers. Across multiple models and tasks the correlation is weak to moderate and cosine similarity misranks layers in the vast majority of cases. Motivated by these findings, the paper advocates using an accuracy-based relevance score for mechanistic interpretability and for structured pruning, and shows that pruning guided by this metric outperforms several established baselines in both task-dependent and task-independent settings, albeit at higher computational cost.

**Strengths:**

The paper addresses an important and timely problem for both interpretability and model compression: how to evaluate which layers actually matter. The theoretical result is clear and convincing in demonstrating the possibility of pathological cases that invalidate cosine-similarity as a universal proxy. The empirical evaluation is broad and carefully presented, spanning multiple model families, many datasets, and both task-dependent and task-independent pruning regimes. The manuscript also provides actionable methodology: a fully defined accuracy-based relevance score, iterative pruning procedures, and cost accounting that clarifies the trade-offs between fidelity and compute. The analyses of variance across tasks, the examples where layer removal improves performance, and the distillation of insights back into pruning recommendations are all valuable for practitioners who want dependable diagnostics instead of relying on a cheap but misleading proxy.

**Weaknesses:**

While compelling, the work leaves several practical and methodological gaps that, if addressed, would strengthen the contribution. First, the accuracy-based metric is expensive. Please quantify wall-clock runtime and monetary cost for representative experiments and provide more detail on how costs scale with model size, number of layers, and calibration set size. Second, task-independent performance is sensitive to the calibration dataset. Please report more systematic ablations that identify which properties of a calibration set make it effective and whether simple ensemble calibration strategies mitigate sensitivity.

**Questions:**

See above.

---

> ### Author Response · Authors · 2025-12-03
>
> We thank the reviewer for the thoughtful and comprehensive assessment of our work. We especially appreciate the recognition that our contribution goes beyond proposing a new pruning strategy: as noted, our results address a central open problem in mechanistic interpretability—how to reliably determine which layers truly matter in transformer models. This broader interpretability significance can be easy to overlook if the paper is viewed solely through the lens of pruning, and we are grateful that the reviewer highlighted it.
>
> We also thank the reviewer for acknowledging key strengths: the clarity of the theoretical result, the breadth and rigor of the empirical evaluation across multiple model families and datasets, and the practical value of the accuracy-based relevance score. Our goal was to provide a principled and dependable alternative to cosine similarity—one that supports both mechanistic interpretability and structured pruning—and we are glad this came through clearly.
>
> The review raises two concerns: (1) Computational cost and (2) Need for a more systematic ablation of the calibration set in the task-independent setting.
>
> ---
>
> **Computational Cost**
>
> In the revised manuscript, we include extensive runtime analysis for all methods and examine scalability to larger models and varying calibration set sizes (Section 6.1, Appendix E.6).
>
> Note that our goal was not to design the fastest metric, but the most faithful one—accurately capturing each block’s contribution to downstream performance. Speed optimizations, while valuable, fall outside the scope of this paper. As such, the newly added runtimes represent upper bounds; our relevance computation is inherently parallelizable across layers, and future work could leverage quantization and optimized inference to reduce runtime substantially.
>
> As stated in the original submission, our method is faster than prior pruning approaches (e.g., perplexity and output-based methods) but slower than cosine similarity, which is extremely fast but—as we show—is unreliable for measuring layer relevance in LLMs.
>
> Mean wall-clock times for pruning 25% of LLaMA3-8B across four datasets (Table 3):
> - Accuracy (ours): 4.6 hours
> - Cosine Similarity: 8.6 minutes
> - Perplexity: 4.62 hours
> - Output Cos-Sim / Norm: 4.8 hours
> - Output Divergence: 4.9 hours
> - Taylor: 10.4 hours
>
> For all models, runtime depends strongly on the calibration dataset. For instance, for LLaMA-3-70B at 50% pruning, the runtime ranges from ~8.5 days with C4 to ~1.1 days with CodeAlpaca (Section 6.1; Appendix E.6). We also show linear scaling with calibration set size (Figure 20).
>
> ---
>
> **Role of Calibration Dataset**
>
> We strengthened our task-independent analysis by adding broader cross-task pruning experiments (Table 2, Section 6; Table 7, Appendix E.5). Results reinforce a key observation: layer relevance is inherently task-dependent, so robust task-independent pruning may require calibration sets with diverse linguistic and reasoning properties.
>
> Notable findings:
> - MMLU and BoolQ consistently fail to be calibrated by other tasks.
> - For MMLU, we hypothesize that its broad coverage of world-knowledge domains requires a calibration set with similar topical diversity.
> - For BoolQ, its unique yes/no question format—absent from other datasets—limits transferability.
>
> These observations suggest that calibration-set composition plays a critical role in generalization. While our interpretations are preliminary, they highlight an important direction for future work: systematically characterizing which properties enable effective task-independent pruning.

---

### Author Response · Authors · 2025-11-28
**We thank the reviewers for their feedback and present a revised manuscript that strengthens our core contribution**

We thank the reviewers for their positive feedback and thoughtful comments. We are excited about the research directions our work opens—both for mechanistic interpretability and structured pruning. Our paper challenges the widespread use of cosine similarity as a proxy for layer relevance in LLMs, showing through theory and experiments that it often misrepresents true importance. To address this, we propose a simple yet effective alternative: measuring relevance in terms of accuracy. By replicating prior interpretability experiments with our accuracy-based score, we uncover substantially different insights into LLM layer relevance. Furthermore, pruning with our metric achieves state-of-the-art results in structured pruning.

We have uploaded an improved version of the manuscript following the reviewers' recommendations, with major changes highlighted in blue. Key updates include:

1. **Task-Independent Performance**: We expanded our discussion on pruning with different calibration sets. Using only one task’s training set, we can prune and generalize well to other tasks—though layer relevance remains inherently task-dependent. High task-independent performance likely requires a calibration set with diverse tasks.

2. **Computational Costs**: We now report wall-clock times for pruning 25% of LLaMA3-8B across four datasets using all methods in Section 6 on the same GPU model. We also provide worst-case estimates for pruning 50% of LLaMA3-70B with our current (suboptimal) implementation. For example, pruning with C4 would take 8.5 days, while CodeAlpaca would take 1.1 days.

3. **Search-Based Pruning**: As noted by reviewer BfoK30, our results could potentially improve by identifying the optimal combination of layers to prune according to our metric. However, achieving this would require a combinatorial search (e.g., A* or exhaustive enumeration), which is currently computationally infeasible. For example, pruning just three layers from a 32-layer model involves evaluating tens of thousands of combinations.

4. **Healing Results**: Following reviewer a5Py28’s suggestion, we added results with healing—a post-pruning fine-tuning phase. Importantly, our accuracy-based method still outperforms cosine similarity under healing, though the gap depends on pruning ratio:
    - At 25% pruning: accuracy+healing $\approx$ 60.5% vs. cosine+healing $\approx$ 59.6%.
    - At 50% pruning: accuracy+healing $\approx$ 42.2% vs. cosine+healing $\approx$ 31.2%.
    - At 75% pruning: all methods degrade severely, but accuracy+healing remains slightly better (12.2% vs. 9.3%).

We believe these revisions address the main concerns and strengthen our contributions.

Thank you for your constructive feedback—we hope this updated version raises your opinion of our work.

---

### Author Response · Authors · 2025-12-03
**TL;DR for AC**

**Reception**: Reviewers praised the paper’s contribution, clarity, and motivation.

**Weaknesses**: Requests for additional experiments—not methodology or correctness. We addressed these with substantial new experiments (hence the delayed response).

**Revisions**: Minor textual edits (highlighted in blue) + major new empirical results that strengthen the paper without changing its original conclusions.

**Score implications**: Two reviewers explicitly indicated willingness to raise scores if concerns were resolved; we believe they are fully addressed.

**Core contribution**:
1. Show that cosine similarity—widely used for layer relevance in LLM interpretability and pruning—is unreliable.
2. Provide theoretical and empirical evidence of its weaknesses.
3. Propose a robust accuracy-based metric; re-evaluate prior work and show previous conclusions were misleading.
4. Demonstrate our metric achieves best performance across diverse pruning setups (task-dependent/independent, iterative, one-shot, with/without healing).

**Main concerns addressed**:
1. **Wall-clock time**: Added runtime measurements; our method is second-fastest after cosine similarity.
2. **Task-independent setup**: Added detailed discussion and conditions for reliable use.
3. **Post-processing (healing)**: Added new experiments showing our method performs best here too.

---

### Meta-Review · Area_Chair_d6t1 · 2026-01-11

**Summary:**

This paper proposes an accuracy-driven metric for eliminating layers of an LLM towards model compression. Previous work used cosine similarities between a layer's input and output, while this work shows that cosine metric is not a faithful metric since there are tasks where an important layer can be removed which has high cosine similarity. The proposed approach simply finds the layer whose removal minimally impacts the performance on a calibration set. They find this pruning is very task-dependent unlike previous results.

Main concerns raised by reviewers were on complexity of pruning, and the task dependent nature of calibration set. Authors have added wall clock time showing their approach is faster than other pruning approach except cosine similarity but which doesn't perform well, and they show there is generalization across tasks. I think these results address the reviewers concern to some extent.

Further, there is some novelty concerns here. E.g., rank-reduction pruning approach such as LASER (Sharma et al. 2024), also uses a task-dependent calibration set and filters based on accuracy. LayerSkip is another related work that is not cited and discussed (Elhoushi, et al., 2024). Authors should differentiate their approach more.

However, the result here is simple and potentially useful and so I am going with an accept.

**Reviewer Concerns:**

1. Reviewer J9R4 raised concerns on computational complexity and details of calibration data. Authors present results that shows that this calibration metric does take significant time, and that using a general-purpose calibration data can hurt performance. I think this was expected and is okay.

2.  Reviewer BfoK raised concerns about the search procedure, task-dependent nature of calibration, and asked for wall clock time. Authors have added wall clock time, argued that the proposed search procedure is designed to keep it tractable even if it is not most efficient, and experiments show that there is some generalization across tasks.

3. Reviewer a5Py also raised concerns about computational feasibility, along with asking whether layer-relevance can be due to simply model not adjusting to some norm changes in input, rather than intrinsic 'knowledge' in the layer. Authors show that their approach is faster than all approaches except cosine but which performs worse. Further, they ran new experiment with healing and show their approach continues to outperform cosine. While this maybe true, it will be great to see what layers are selected with and without healing for the proposed approaches and the downstream performance difference.

4. Reviewer 5sPj also raised concerns about computational feasibility, gap between theory-practice, and whether task knowledge leaking through calibration set can give the proposed approach an advantage over cosine similarity. While the proposed construction in Theorem 1 is contrived, I think it will be interesting for folks working on theory of transformer. Authors also have shown that pruning with one task can show improvement on other tasks. Authors have added wall clock time as well showing their approach is faster than all except cosine similarity.

**Reviewer Scores:**

1. Reviewer J9R4 will likely keep their score at 6.

2. Reviewer BfoK may have increased their score to 6 since some of their concerns (e.g., wall clock details) have been addressed.

3. Reviewer a5Py may have increased their score to 6 given new experimental results.

4. Reviewer 5sPj may have increased their score to 6.

Overall, this paper is tending towards a weak accept.

---

### Decision · Program_Chairs · 2026-01-26

Accept (Poster)